# SSolar-GOA v1.0: a simple, fast, and accurate Spectral SOLAR radiative transfer model for clear skies

Victoria Eugenia Cachorro[1], Juan Carlos Antuña-Sanchez[1], and Ángel Máximo de Frutos[1]

[1]Group of Atmospheric Optics, Universidad de Valladolid (GOA-UVa), Valladolid, 47011, Spain

*Correspondence to*: Victoria E. Cachorro (chiqui@goa.uva.es)

**Abstract.** The aim of this work is to describe the features and to validate a simple, fast, accurate and physically-based spectral radiative transfer model in the solar wavelength range under clear skies. The model, named SSolar-GOA (the first "S" stands for "Spectral"), was developed to evaluate the instantaneous values of spectral solar irradiances at ground or at the bottom
surface level. The model requirements are designed based on the simplicity of the analytical expressions for the transmittance functions in order to be easily replicated and applied by a wide community of users for many different applications (atmospheric and environmental research studies, satellite remote sensing, solar energy, agronomy/forestry, ecology, and others). Although spectral, the model runs quickly and has sufficient accuracy for the evaluation of solar irradiances with a spectral resolution of 1-10 nm. The model assumes a single mixed molecule-aerosol scattering layer where the original
Ambartsumian method of "adding layers" in a one-dimensional medium is applied, obtaining a parameterized expression for the total transmittance of scattering. Absorption by the different atmospheric gases follows "band model" parameterized expressions. The input parameters must be realistic and easily available since the spectral aerosol optical depth (AOD) is the main driver of the model. The validation of the SSolar-GOA model has been carried out through comparison with simulated irradiance data from the libRadtran package and with direct/global spectra measured by spectroradiometers. Thousands of
spectra under clear skies have been compared for different atmospheric conditions and solar zenith angles (SZA). The SSolar-GOA is validated by a quantitative comparison with libRadtran, showing that it underestimate direct normal, global, and diffuse spectral components with relative differences of +1% (RMSE%=4.6-8), +3% (RMSE%=5.3-8), and 8% (RMSE%=9.3-9.6), respectively, when the SZA varies from 6º to 60º. Compared with the measured irradiance data of the LI-1800 and ASD spectroradiometers, the relative differences of direct normal and global components are within the overall experimental error,
about ±2-12% (RMSE%=5-8.3), with underestimated or overestimated values. The diffuse component presents the highest degree of relative difference that can reach ±20-30% and RMSE of 25-50%. The relative differences depend strongly on the spectral solar region analysed and the SZA, but the high values of RMSE are due to the artifice generated by the different spectral resolution of the absorption coefficients of both models. Model approach errors combined with calibration instrument errors may explain the observed differences. The SSolar-GOA v1.0 is implemented in Python and open-source licensing.

# 1 Introduction

Solar radiation is the primary energy source of the Earth-atmosphere system. It is the driver of the most important mechanisms of the atmospheric/climate system, mainly through radiation energy balance and the greenhouse effect (Goody, 1964; Houghton, 2002; Wild et al., 2013). Solar radiation governs thermal and hydrological conditions which are fundamental for life on Earth, as well as the environment, ecology, agriculture, forestry, etc. Today, solar radiation is also of great importance in other areas, i.e., solar energy, urban building design, engineering applications. Therefore, measurements and modelling of solar radiation are essential in many fields. The evaluation of global, direct, and diffuse components is of particular importance. Earth surface solar radiation measurements are currently carried out using broadband radiometers at meteorological stations from different national weather services or more specific worldwide radiometric networks, such as the Global Radiation and Aerosols (GRAD, 2021) of the ESRL Global Monitoring Laboratory-NOAA, or the National Solar Radiation Database (NSRDB-NREL, 2021). The diversity of solar radiation networks with different objectives and applications presents variable data quality; only specific networks can guarantee the quality of solar radiation data, such as the BSRN (BSRN: Baseline Surface Radiation Network, 2021), to ensure climatological trend studies or precise values for global balance in the Earth system (Wild et al., 2013; Wild, 2009).

This work focuses on spectral solar surface radiation measurements which give continuous spectra for a wide spectral range (i.e., UV (~300-400 nm), visible (~400-700 nm), near-infrared (~700-1000 nm), entire solar range (~300-3000 nm)) under clear skies. Broadband solar radiation data are very abundant, but spectral solar radiation measurements are comparatively scarce. Generally, well-established networks are not available for this purpose, and most of the known spectral solar data are restricted to specific research campaigns, although some research centres and research groups have recorded important databases: NREL Spectral Solar Radiation Data Base, 2021; GOA-UVA solar radiation, 2021; WOUDC, the World Ozone and Ultraviolet Radiation Data Center. The main reason for this is that the instruments for these measurements – the spectroradiometers – are more complex electro-optical systems for field measurements, and calibration procedures and maintenance are difficult to perform routinely in a non-operational network. One example is the MFRSR (Hodges, 1993) USA network, which provides spectral radiation data but only at specific wavelengths. Today, well-known CCD arrays-based detection systems are part of modern spectroradiometers, which are increasingly used, facilitating spectral measurements.

However, it is possible to find many references in the literature which are focused on instruments, measurements, and modelling of surface spectral solar radiation (Leckner (1978); Koepke and Quenzel (1978); Bird (1984); Cachorro et al., (1985, 1987a, b, c, 1997); Bird and Riordan (1986); Riordan et al., (1989); Gueymard (1995, 2001, 2005, 2008, 2019); Utrillas et al., (1998, 2000); Kiedron et al., (1999); Mlawer et al., (2000); Martínez-Lozano et al., (2003); Bais et al., (2005); Michalsky et al., (2006); Habte et al., (2014); Egli et al., (2016); Mlawer and Turner (2016)). These types of surface spectral solar measurements are also extensively used to retrieve the content and properties of different atmospheric components such as water vapour, ozone, aerosols, etc. (Cachorro et al., 1986, 1996, 1998, 2000a, b; Martínez-Lozano et al., 1998; Carlund et al., 2003; Vergaz et al., 2005; Toledano et al., 2006; Estellés et al., 2006). Although atmospheric/climate sciences and solar energy

are the most important fields where spectral solar radiation data are required, other fields also apply them, as can be seen in a recent publication of Gueymard (2019). Spectral solar radiation data are currently in great demand by the photovoltaic (PV) community for solar power due to the extensive use of PV modules whose performance must be evaluated (Norton et al., 2015; Amillo et al., 2015; Sengupta et al., 2018).

Spectral solar radiation measurements have been carried out by the "Grupo de Optica Atmosférica" of the "Universidad de Valladolid, (GOA-UVA)" for more than two decades in conjunction with the development and use of different solar radiation models, as part of its routine work in atmospheric studies and in other related areas (Cachorro et al., 1985, 1987a,b,c, 1997, 1998; Vergaz et al., 2005; Toledano et al., 2006; Berjón et al., 2013). The modelling of these measurements is the main aim of this work: to set-up and to validate a simple, fast, and accurate spectral solar radiative transfer model covering the entire solar range. The model is especially suited for the measurements of spectroradiometers working in low to medium spectral resolution (i.e., 1-10 nm). The idea is to provide a radiation spectral model to a wide community of users; thus, the model must be theoretically simple and easy to use and replicate. Fast calculations of the model are devoted especially for network-routine data of high-time resolution, long data series analysis or reconstruction, satellite solar radiation estimation, and applications in solar energy or other areas. The SSolar-GOA is now as v1.0, implemented in Python and open-source licensing.

The paper is structured as follows: Section 2 briefly describes the characteristics of the two spectroradiometers employed to perform the solar spectral measurements and gives a general theoretical background in the context of solar radiation modelling. Section 3 describes the SSolar-GOA model. Section 4 presents the results of validation of the SSolar-GOA model by the comparison with libRadtran package (libRadtran User´s Guide 2015, 2020) which was used as the benchmark, and also by the comparison with experimental solar spectral radiation data. Conclusions and recommendations are also discussed in the last section.

## 2 Material and methodology

### 2.1 Instrumentation and measurements

The experimental measurements of solar spectral irradiance for the validation process of the SSolar-GOA model were taken using two commercial spectroradiometers. The first spectroradiometer used was the LI-1800 model from Li-COR Biosciences (LI-COR, 1989), which covers the 300-1100 nm spectral range and is based on monochromator holographic grating of 800 grooves/mm with a nominal FWHM (Full Width at Half Maximum) or spectral resolution (s.r.) of 6 nm (according to Vergaz et al. (2000) the FWHM measured at our Laboratory was 6.25 ±0.07 at 632.8 nm He-Ne laser wavelength). The scanning system of LI-1800 takes about 40 seconds to measure a solar spectrum. The software of the system allow variable wavelength sampling, but currently 1 nm and also a programmable time are used for the measurement of global solar radiation spectra (or the direct component with a solar tracker). The LI-1800 is manufactured with a Remote Cosine Receptor for global solar irradiance measurements, but different fore-optic devices designed by the GOA Group allows for direct normal irradiance and reflected solar irradiance measurements (Durán, 1997).

The other spectroradiometer used was the FieldSpec Pro (hereafter, ASD), a general purpose portable spectroradiometer developed by ASD Inc. (ASD Full Range, Portable Spectrometers & Spectroradiometers | Malvern Panalytical, 2021; Milton et al., 2009; Goetz, 2012; Hannula et al., 2020). This spectroradiometer covers the 350-2500 nm shortwave range and is composed of three spectrometers: the VNIR from 350-1050 nm is composed of a 512-channel silicon photodiode array (CCD) overlaid with an order separation filter, a second scanning spectrometer (SWIR-1) from 1050 nm to 1800 nm, and a third scanning spectrometer (SWIR-2) to 2500nm. Each SWIR consists of a concave holographic grating and a single thermoelectric cooled indium gallium arsenide (InGaAs) detector. Each grating is mounted on a common shaft which oscillates at 100 ms/scan, thus providing their spectra in a few seconds and the CCD array makes it simultaneously in the VNIR spectral range. The spectral resolution of the ASD is different for each of the three spectrometers: the VNIR has approximately 3 nm of s.r. at around 700 nm and the SWIRs have about 10-12 nm.

The ASD system is provided with specific fore-optical accessories for field radiance and irradiance measurements of different FOVs (Field of View of 1, 3, and 8 degrees), and a Remote Cosine Receptor is used for global irradiance and for measuring full-hemisphere albedo or reflectance spectra. Light collection is achieved through a bundle of optical fiber. For direct normal irradiance measurements, the earlier fore-optic accessories used for radiance measurements cannot be used. This is because each tube is provided with a lens which focuses the radiation over the optical fiber, and due to the high energy of the normal direct irradiance, this may damage the fiber. Thus, a new tube collimator was designed by the GOA Group which can be used with the ASD.

Calibration details, associated errors, and measurements of the LI-1800 for both direct and global irradiances were discussed in Vergaz (2001); Martínez-Lozano et al. (2003); Vergaz et al. (2005); Estellés et al. (2006). As a general feature, the LI-1800 presents an experimental error of about 5% into the 340-1100 nm spectral range while the instrument itself has proven to be durable and with a long-lasting calibration. The ASD solar irradiance measurements have similar errors to LI-1800, although the advantage of registering near instantaneous spectra and automatic optimization, the latter results in variable integration times and gains for one measured spectrum over another, being that a drawback in the data processing. This requires special care and attention for field solar irradiance measurements, and normally post-processing is necessary since frequent saturation is observed in the spectra which is not always avoidable. In both spectroradiometers, the LI-1800 and ASD diffuse solar irradiances are derived from the difference between near simultaneous measured spectra of global and direct normal irradiances.

**2.2 General theoretical background for solar spectral irradiance models at surface level**

The global solar spectral irradiance GHI(SZA, λ) at ground level over a horizontal surface and for a given sun position (specified by the Solar Zenith Angle, SZA) can be expressed as the sum of its direct normal component (DNI(SZA, λ)) projected onto the horizontal surface (hence multiplied by cos (SZA)), plus the horizontal diffuse irradiance component DIF(SZA, λ), also dependent on the SZA.

$$GHI(SZA, \lambda) = DNI(SZA, \lambda)cos(SZA) + DIF(SZA, \lambda) , \qquad (1)$$

Although the wavelength is explicit in the above expression (1), it should be noted that it is valid for both spectral and integrated irradiance values (in this case removing $\lambda$ and considering the integration over the entire solar range). These quantities are usually expressed in the units of $Wm^{-2} \mu m^{-1}$ ($Wm^{-2} nm^{-1}$) for spectral irradiance values, and $Wm^{-2}$ for integrated irradiance values. There is not a unified nomenclature to designate the three components of solar radiation at surface level, (global or total horizontal solar irradiance (GHI) is also called shortwave downwelling solar irradiance (SWD), shortwave surface

irradiance (SSI), surface total solar flux, etc.). Therefore, expression (1) incorporates the most recent and most widely used names in the solar energy community for these irradiances.

If we divide these irradiances by the irradiance at the top of the atmosphere (the extraterrestrial irradiance, $F_o$, multiplied by the corresponding correction of the Earth-sun distance (D) projected over the horizontal plane, $D*F_o cos(SZA)$), the corresponding atmospheric transmittances at surface level for the above three components are obtained: global transmittance

$T_{GHI}(SZA, \lambda)$, normal direct transmittance $T_{DNI}(SZA, \lambda)$, and diffuse transmittance, $t_{DIF}(SZA, \lambda)$.

$$T_{GHI}(SZA, \lambda) = T_{DNI}(SZA, \lambda) + t_{DIF}(SZA, \lambda), \qquad (2)$$

Here it must be noted that using the transmittances in expression (2), the horizontal global transmittance ($T_{GHI}$) is given by the

145 sum of the normal direct transmittance ($T_{DNI}$) and the diffuse horizontal transmittance ($t_{DIF}$). As can be seen, the explicit dependence on the cos(SZA) of expression (1) is removed in expression (2). The advantage of using transmittance functions instead of irradiance values is because in this way it works with normalized functions whose values are always equal to or less than 1. Besides, expression (2) is also valid for integrated values of solar radiation, which translates to the definitions of the clearness indices Kn and $K_T$ for normal direct and global solar components, respectively, where Kn=$T_{DNI}$ and $K_T$=$T_{GHI}$, in this

case referring to instantaneous and integrated values. Therefore, expression (2) is now written as Kn= $K_T$(1-K), where K is the fraction of the diffuse radiation (K=DIF/GHI). These indices are widely used by the solar energy community, and are the base of the so-called separation solar radiation models under all sky conditions (Gueymard and Ruiz-Arias, 2016; Yang and Boland, 2019).

The direct normal spectral solar component at any level of the atmosphere (expressed as radiance or irradiance quantities) is

155 currently given by the Beer-Lambert-Bouguer (BLB) law. This law is a solution of the Radiative Transfer Equation (RTE) when applied only to direct component. The simplicity of the resulting solution makes it possible to consider scattering by molecules-particles, and absorption by atmospheric gases as independent processes (non-interaction between them). This allows to present $T_{DNI}$ as a product of independent transmittances of the different atmospheric constituents: ozone, water vapour, aerosols, molecules, etc. (see paragraph 3.1).

Therefore, it is standard in Radiative Transfer Theory to separate the modelling of solar radiation into its two components, direct normal and diffuse, and solving the RTE for each component, considering a dispersive or scattering medium without absorption of atmospheric gases. However, solving the RTE for the diffuse component is not a straightforward task and different analytical and numerical methods have been developed depending on the approaches or the specific problem involved (see classical books on Radiative Transfer Theory: Chandrasekhar, 1960; Sobolev, 1963; Kondratyev, 1969; Lenoble, 1985,

1993; Liou, 1992, 2002; Zdunkowski et al., 2007; Kokhanovsky, 2008; see also the different solvers used in libRadtran).

Since we are interested in solar spectral irradiances or fluxes and not radiances a more convenient approach for solving the RTE is addressed by those methods known as "two streams" or "two flux" which indistinctly solve the RTE for the diffuse component only or for the global component. The "two flux" methodology was extensively developed in the 70-80s and presents numerous variants (Joseph et al., 1976; Meador and Weaver, 1980; Zdunkowski et al., 1980; King and Harshvardhan,

1986; Liou, 1992; Fouquart and Bonnel, 1980; Durán, 1997; Räisänen, 2002, Lin et al., 2019).

Although less frequent, another possible option is to consider other methods, such as the original method of "addition of layers" in a one-dimensional scattering medium, developed by Ambartsumian (Sobolev, 1963; Nikoghossian, 2009), which does not consider the RTE. The analytical expression obtained for the transmittance of the global solar irradiance in a scattering medium (without atmospheric gas absorption) composed of aerosols or molecules (or a mixture of both components) is the core of the

175 SSolar-GOA model (see Section 3.2). After that, this transmittance is multiplied by the absorption transmittance of atmospheric gases giving the above spectral $T_{GHI}$ of expression (2) assuming non-interaction between scattering and gas-absorption.

**2.3 The libRadtran package**

The libRadtran package is a software library for radiative transfer calculations of solar and thermal radiation (from 120 nm to 100 μm) in the Earth's atmosphere. The central part of the software package is an executable program called uvspec which was

180 initially developed for UV radiation evaluation and which has undergone numerous extension and improvements to reach the current libRadtran estructure (Mayer and Kylling, 2005; Emde et al., 2016). It is freely available at the web page http://www.libradtran.org, which contains all the available information about the program including the user's guides (libRadtran user´s guide, (2015) and (2020)) and the software source code. The libRadtran package contains a complete treatment of the atmospheric absorption-scattering processes and offers many options for inputs, utilities, methods, and outputs

to handle the complex structure that Radiative Transfer models have, allowing for the determination of the field radiation (radiances, irradiances, polarization, etc.) in the atmosphere. Therefore, libRadtran is a set of RT Codes which serves as a reference tool which is widely used by the scientific community in different fields of study.

libRadtran requires detailed information specified in input files provided by the same package or constructed by the users, for example the Mie program (see libRadtran user´s guide Chapter 4) for the calculation of aerosol optical properties. For the

190 irradiance values, the direct normal component is calculated based on the BLB in a similar way as SSolar-GOA (described in next section 3.1). For our simulations, the algorithm for the spectral solar diffuse horizontal irradiance used sdisort RTE solver with 10 streams. The global spectral radiation is constructed by the sum of direct horizontal plus diffuse horizontal components.

All the atmospheric gases were considered in the libRadtran for the simulations. To compare with SSolar-GOA model, the adequate option of libRadtran are the "spectrally resolved calculation" for the UV and visible spectral range and the "pseudo-spectral" in the infrared solar region (i.e.: water vapour, oxygen and carbon dioxide), represented by the band parameterization of the LOWTRAN7 Code taken by the SBDART model and adopted in libRadtran (Kneizys, 1988; Mayer and Kylling, 2005). Therefore, the latter option was taken by us in accordance with the building of the SSolar-GOA model.

A midlatitude summer atmospheric profile with a default of an aerosol profile in the summer season was chosen, but the contribution of aerosol was constructed on the alpha and beta Ångström turbidity parameters (they are also represented in the text by the symbols α and β, respectively; see next sections). It must be noted that under clear skies the aerosol contribution is the most important factor for solar irradiance and the spectral behaviour of $AOD(\lambda)$ is given by the alpha parameter. The spectral $AOD(\lambda)$ is the most relevant input for a proper comparison between libRadtran and our model since it determines the curvature shape and height of the transmittance of the direct normal component. The other two aerosol parameters, the asymmetry parameter (g) and single scattering albedo (SSA), are of secondary importance and are taken as fixed values (not wavelength dependent) in libRadtran and SSolar-GOA models.

## 3 Description of the SSolar-GOA radiative transfer model

The SSolar-GOA model is designed based on our previous experience gained through using simple empirical parametric spectral solar radiation models (Cachorro et al., 1985) and more complex radiative transfer codes (Cachorro et al., 1997) in an attempt to cover the gap between these two extreme configurations. This is a physical, fast, efficient, and accurate spectral radiative transfer model to estimate the spectral components of solar radiation at surface level or at a given altitude considered as the bottom surface in the model and it covers the solar spectral range, from 300 to 2600 nm. The crux of the model is the simple analytical parameterized expression for the spectral scattering transmittance function of the mixed layer of molecules and aerosols. This expression was developed by Ambartsumian (Sobolev, 1963; Nikoghossian, 2009) for a one-dimensional scattering medium. The atmosphere is assumed to be a single homogeneous plane parallel layer. Absorption by atmospheric gases is given by the parameterized transmittances based on "band model approach" (Pierluissi and Maragoudakis, 1986; Pierluissi and Tsai, 1987; Pierluissi et al., 1989) which were applied to the LOWTRAN7 Code (Kneizys, 1988).

The model presents a moderate spectral resolution, aimet at operation of 1-10 nm, depending on the selected spectral resolution of the extra-terrestrial solar spectra in combination with that of the absorption coefficients of the absorbing gases. The accuracy of the model is in consonance with the error associated with experimental data of the most common commercial spectroradiometers, about 2-5%. Below, we present a detailed description of the SSolar-GOA model, first to evaluate the direct normal component and then the global spectral irradiance, both as independent components. The diffuse spectral irradiance is derived from the other two quantities. The model may be easily adapted to the case of limited available information about model's input parameters. The SSolar-GOA v1.0 is released as free and open-source software. It is implemented in Python offering portability across architectures and operating systems. For download instructions, see Code Availability section.

## 3.1 The spectral direct normal solar irradiance

Assuming the validity of the BLB law, the spectral irradiance of the direct normal component of solar radiation, DNI (SZA, z), at any time (given by the SZA) and at any vertical altitude z of the atmosphere, is given by:

$$DNI(SZA, z, \lambda) = DF_o(\lambda)T_{DNI}(SZA, z, \lambda) = DF_o exp(-\tau(z, \lambda)m), \tag{3}$$

where $\tau(z, \lambda)$ is the spectral atmospheric optical thickness at the level z, or the altitude in the atmosphere which accounts for scattering by molecules and particles as well as absorption by atmospheric gases. $F_o(\lambda)$ is the spectral irradiance at the top of atmosphere (extraterrestrial spectrum) and D is the correction factor of the Earth-Sun distance.

Considering the ground surface level z=0, $\tau(z=0, \lambda)=\tau(\lambda)$ represents the total spectral optical thickness of the atmosphere at the site. m is the relative optical air mass giving the slant path of sun's rays relative to the zenith, which is given by m=1/cos(SZA) for a plane-parallel atmosphere. For a spherical atmosphere, more accurate expressions for m are necessary when the SZA is greater than 70°, in order to account for the curvature of the atmosphere and the refraction effects, various expressions were developed for each atmospheric component (Kasten and Young, 1989; Gueymard, 1995, 2005; Tomasi et al., 1998; Chiron de la Casinière and Cachorro Revilla, 2008; Rapp-Arrarás and Domingo-Santos, 2008).

As mentioned, the advantage of solving the RTE only for the direct normal component is that $T_{DNI}$ can be calculated as a product of transmittances due to the different processes of attenuation due to the different atmospheric components, where the non-interaction between these processes is implicitly assumed.

$$T_{DNI}(SZA, \lambda) = T_R(SZA, \lambda)T_a(SZA, \lambda)T_{gas,i}(SZA, \lambda), \tag{4}$$

In expression (4), the different transmittances are given by exponential functions of the optical thickness of each process: the scattering by molecules or Rayleigh scattering, $\tau_R(\lambda)$, scattering by aerosols, $\tau_a(\lambda)$ or AOD($\lambda$), and the absorption by atmospheric gases, $\tau_{gas,i}(\lambda)$ (subscript i refers to different selected gases), multiplied by the corresponding relative air mass.

$$T_{DNI}(SZA, \lambda) = exp(-\tau(\lambda)m) = exp(-\tau_R(\lambda)m_R)exp(-\tau_a(\lambda)m_a)exp(-\tau_{gas,i}(\lambda)m_{gas,i}), \tag{5}$$

The BLB law of expression (3) is also valid for integrated irradiance values (i.e. removing the wavelength dependency), where $\tau$ represents the integrated total optical thickness of the atmosphere, but expressions (4-5) are only valid for spectral values (Cachorro et al., 2000b, a; Utrillas et al., 2000). Despite this, expression (4) is taken as a good approach for the "Broadband

Solar Models" (Gueymard, 2008; Ruiz-Arias and Gueymard, 2018) assuming that the transmittance of each atmospheric component is an integrated value over the entire solar range. According to (5), it follows that:

$$\tau(\lambda)m = \tau_R(\lambda)m_r + \tau_a(\lambda)m_a + \tau_{gas,i}(\lambda)m_{gas,i},\qquad(6)$$

Therefore, the total optical thickness of the atmosphere is given by the sum of the different optical thicknesses due to the different attenuation processes of solar radiation assuming the same relative optical air mass. According to expressions (5) and (6) we can use either transmittances or optical thickness. Observe that optical thickness is a dimensionless parameter like the relative optical air mass. Although it is usual to consider $m=m_R=m_a$, for atmospheric gases it is more convenient to use different expressions specifically determined for each absorbing gas (Gueymard, 1995; Tomasi et al., 1998).

There are different parameterized expressions to evaluate the Rayleigh optical thickness (Teillet, 1990; Gueymard, 1995; Bodhaine et al., 1999; Tomasi et al., 2005) with insignificant differences for our purpose, and hence the Gueymard (1995) formula was taken for the SSolar-GOA model:

$$\tau_R(\lambda) = \frac{1}{117.2594\lambda^4 - 1.3215\lambda^2 + 0.00032 - 0.000076\lambda^{-2}},\qquad(7)$$

Since this expression is evaluated at sea level, it is necessary to multiply by the factors P and Po, where P and Po are the pressure at the site (or altitude) and the sea-level pressure, respectively. The transmittance of scattering by aerosols $T_a$ (SZA) is accounted for by a simple approach for the aerosol optical depth given by the Ångström formula (Ångström, 1929, 1930, 1961; 1964). This is an empirical expression extensively used in the field of aerosol studies and in solar radiation applications

(Cachorro et al., 1987b, c, 2000a, b) and is expressed as follows:

$$\tau_a(\lambda) = \beta\lambda^{-\alpha},\qquad(8)$$

where α (alpha) and β (beta) are the Ångström turbidity parameters. The α parameter, also called Ångström exponent (the

280 symbol AE is now more commonly used in place of α), is related to the bulk size of the particles, and the β parameter is the aerosol optical thickness at 1 µm wavelength. Bear in mind that this is an empirical expression which may be applied to a given extended spectral range. Frequently, two wavelengths can be selected and hence expression (9) allows for the determination of the α parameter:

$\dfrac{\tau_a(\lambda_1)}{\tau_a(\lambda_2)} = \left(\dfrac{\lambda_1}{\lambda_2}\right)^{-\alpha},$ (9)

When several wavelengths are available, such as in sun-photometers or spectroradiometers, the α-β parameters can be determined simultaneously by a linear regression of $\log[\tau_a(\lambda)]$ versus $\log[\lambda]$ (Cachorro et al., 1987b, c, 1989, 2000b; Martínez-Lozano et al., 1998). In this case, different values of the α-β pair are obtained depending of the selected spectral range (or wavelengths). Therefore, some solar radiation models use more than one pair of α-β values to cover the entire solar spectral range (Gueymard and Myers, 2008). However, in the SSolar model, only a pair of values is taken. Despite its simplicity, the Ångström formula has proven to be an excellent approach for modelling the spectral behaviour of the aerosol optical depth, $AOD(\lambda)$. In RT studies, the aerosol optical thickness and other optical properties are determined by the Mie scattering theory (Bohren and Huffman, 1998; Cachorro and Salcedo, 1991). In Cachorro et al. (2000a), experimental direct normal irradiance measurements of the LI-1800 together with rigorous Mie scattering expressions were used to determine the distribution of aerosol particle size and other aerosol parameters.

The absorption processes by atmospheric gases must be accounted for and are given by different transmittances. In the solar spectral range, the SSolar-GOA model uses tabulated absorption coefficients of water vapour ($H_2O(v)$), ozone ($O_3$), oxygen ($O_2$), nitrogen dioxide ($NO_2$), and carbon dioxide ($CO_2$), but the current version only considers water vapour, ozone, and oxygen because of the low absorption-features of the other two components and the necessity of a rapid running of the model. These absorbing gases are represented by the product of the different transmittances.

$$T_{gas,i}(SZA,\lambda) = T_{H_2O}(SZA,\lambda)T_{O_3}(SZA,\lambda)T_{O_2}(SZA,\lambda), \tag{10}$$

The selective line absorption of these molecular gases is treated under the "band model approach" method mentioned above. This results in parameterized expressions which are adequate for models of low-median spectral resolution, as explained below. The transmittance of ozone absorption is given by the expression:

$$T_{O_3} = exp\left(-\tau_{O_3}(\lambda)m_{O_3}\right) = exp\left(-C_{O_3}(\lambda)L_{O_3}m_{O_3}\right), \tag{11}$$

where $\tau_{O3}(\lambda)$ is the spectral ozone optical thickness. $C_{O3}(\lambda)$ refers to the ozone absorption coefficients (or cross-section, depending on the units taken), which carry the wavelength dependence, and $L_{O3}$ is the columnar ozone content. Usually, $L_{O3}$ is given in Dobson units, DU (1 DU=1atm-cm *$10^{-3}$), and thus the absorption coefficients are given in (atm-cm)$^{-1}$. The relative optical air mass, $m_{O3}$, is given by the expression from Komhyr, (1980). Ozone in the region of 280-350 nm corresponds to the Hartley (200-310 nm) and Huggins (300-350 nm) bands, and the Chappuis band in the visible range (400-650 nm). The cross-sections taken in our model for the UV region are those from Bass and Paur (1985). The original values are given with a spectral resolution of 0.05 nm, so they were convoluted with a triangular slit function of 7 nm of FWHM and evaluated or interpolated in 1 nm steps. The cross-sections were also provided for three different temperatures and 226 K was selected for our model. For the visible Chappuis band, the $C_{O3}$ values were taken from Amoruso et al., (1990), Anderson and Mauersberger,

(1992), and Brion et al., (1998). These values were also accommodated to the spectral resolution as before. These $C_{O3}(\lambda)$ values are sufficient to predict solar irradiance values (Redondas et al., 2014; Orphal et al., 2016).

The transmittance of water vapour is given by the parameterized expression of Pierluissi et al. (1989):

$$T_{H_2O}(\lambda) = exp(-\tau_{H_2O}(\lambda)m_{H_2O}) = exp\left[\left(-C_{H_2O}(\lambda)Wm_{H_2O}\right)^a\right], \tag{12}$$

where $C_{H2O}(\lambda)$ refers to the absorption coefficient of water vapour which was taken from LOWTRAN7 with a spectral resolution of 20 cm$^{-1}$ in steps of 5 cm$^{-1}$. These coefficients were accommodated as before at a spectral resolution of 7 nm and step of 1 nm. The parameter "a" presents a smooth dependence on wavelength and is given by Pierluissi et al. (1989) for each water absorption band. W is the equivalent absorber amount over the vertical which is related to the amount of absorber, U, or
330 precipitable water vapour, PWV, expressed in cm or gr/cm$^2$ by:

$$W = \left(\frac{P_e}{P_o}\right)^n \left(\frac{T_o}{T_e}\right)^m U, \tag{13}$$

The above expression applied the Curtis-Godson approximation to the whole single layer of the atmosphere for our model,
where Pe and Te are the effective pressure and temperature of the atmosphere, respectively, (we take those of the standard atmosphere) and To and Po are the values at standard conditions. The parameters n and m are also given by Pierluissi et al. (1989) for each band of water vapour. An integration is used to model several atmospheric layers, where Pe and Te are substituted for the values P(z) and T(z), and U by dU=$\rho_v$(z)dz where $\rho_v$(z) is the profile of water vapour density.

The transmittance of oxygen is treated with an expression similar to that of water vapour (Pierluissi and Maragoudakis, 1986).
In contrast to the variability of water vapour, oxygen is constant in the atmosphere. The value used in our model for the equivalent vertical oxygen content was 87068.53 atm-cm, corresponding to the mid-latitude summer atmosphere. This value does not differ substantially for other atmospheres, and therefore, no variation in transmittance was observed. In all the absorbing gas transmittances, the amount of absorbing gas is given in units of atm-cm or in gr/cm$^2$ and hence the absorption coefficients have the inverse units. As can be seen, the procedure followed for the absorption gas transmittances in our model
is equivalent to the "pseudo-spectral calculations" according to libRadtran.

### 3.2 The total (global) scattering transmittance for a mixed aerosol-molecule atmosphere

The simplicity of the SSolar_GOA model is based on the parameterized expression (14) to calculate the total scattering transmittance $T_{Mix}$ (SZA) for a mixed layer of aerosol and molecules, considering the interaction between the two scattering processes. These expressions (14-16) are obtained by the original method of "addition of layers" in a one-dimensional medium
developed by Ambartsumian (Sobolev, 1963; Nikoghossian, 2009). For simplicity, the wavelength is removed in some of the

next expressions (14-16), but the generic parameters $\tau(\lambda)$, $\omega(\lambda)$, and $g(\lambda)$ carry this wavelength dependence and the following subscripts are: (R) for the molecules, (a) for the aerosols, and (Mix) for the mixture. Expressions (14-16) are as follows:

$$T_{Mix}(SZA) = \frac{(1-r_o^2)exp(-k\tau_t m)}{1-r_o^2 exp(-2k\tau_t m)},$$

(14)

where $\tau_t(\lambda)$ is the total scattering optical depth ($\tau_t = \tau_a + \tau_R$), and m is the relative optical air mass. The parameters $r_o$ and k are given by:

$$r_o = \frac{k-1+\omega_{Mix}}{k+1-\omega_{Mix}} \qquad k = (1-\omega_{Mix})(1-\omega_{Mix}g_{Mix}),$$

(15)

where $\omega_{Mix}$ and $g_{Mix}$ are the single scattering albedo and the asymmetry parameter of the mixed layer of aerosols and molecules defined by the corresponding parameters of individual molecules (R) and aerosols (a). They are given by the expressions:

$$\omega_{Mix} = \frac{\tau_R+\omega_a\tau_a}{\tau_t}, \qquad \text{with } \omega_a \neq 1 \qquad g_{Mix} = \frac{g_a\tau_a}{\tau_t},$$

(16)

However, we must state that the transmittance of expression (14) may be also used to evaluate an isolated aerosol layer, represented by the scattering aerosol transmittance, $T_a$. In this case, we need an expression for the scattering transmittance for an isolated pure Rayleigh atmosphere, $T_R$, for example that given in Vermote and Tanré, (1992). The total transmittance of

370 the scattering atmosphere of the aerosol and molecular mixed layer is obtained as the product $T_{a-R}=T_a T_R$, where it is implicitly assumed that there is no interaction between the molecules and aerosols. Therefore, $T_{a-R}$ is equivalent to $T_{Mix}$, but not the same. No significant differences have been found between these two approaches for moderate atmospheric aerosol loads. Scattering and gas absorption are applied to a single atmospheric homogeneous layer in the SSolar-GOA under the consideration of non-interaction of both processes, which simplify considerably the formulation of the model.

The above expressions were derived assuming a zero reflectance or albedo of the underlying surface (considered as a black body), so its influence must be taken into account by the contribution of the multiple reflections between it and the atmosphere. For this effect, we have followed the formulation of Lenoble (1998), where an amplification factor independent of the SZA is defined as:

$f\_amp(\lambda) = 1/(1 - \rho S(\lambda)),$

(17)

where ρ is the surface albedo taken in the model as a constant value and considered Lambertian. $S(\lambda)$ is the spectral atmospheric albedo of the mixed Rayleigh-aerosol layer, given by the sum of both scattering components (Tanré et al., 1986; Vermote and Tanré, 1992).

$$S(\lambda) = S_R(\lambda) + S_a(\lambda),\tag{18}$$

$$S_R(\lambda) = \frac{\tau_R}{2+\tau_R}\left(1 - exp(-2\tau_R)\right),\tag{19}$$

$$S_a(\lambda) = \frac{g'\tau_a}{2+g'\tau_a}\left(1 - exp(-g'\tau_a)\right), \qquad \text{with } g'=\omega_a(1-g_a)\tag{20}$$

As previously mentioned, all expressions (14-20) are wavelength dependent by means of the corresponding parameters. However, due to the difficulty of providing accurate spectral values for the aerosol single scattering albedo $\omega_a$ (or SSA) and the asymmetry parameter $g_a$, these two parameters are taken as constant values in the SSolar-GOA model. These values for the different types of aerosols are given in different publications (Dubovik et al., 2004; Hamill et al., 2016). Finally, we call

attention to the total number of expressions/formulas which define the SSolar-GOA model in comparison with other spectral models of similar characteristics given in the bibliography (e.g.: Bird, 1984; Gueymard, 1995, 2005; Xie and Sengupta, 2018), which results in a major complexity and computational cost.

**3.3 The model input parameters**

According to the above expressions, the input parameters for the SSolar-GOA model are:

  • the solar zenith angle, SZA (degrees)
  • the Julian day, N
  • the pressure at the site, P (in mbar)
  • the surface albedo, ρ

and the aerosol dimensionless parameters are:

  • alpha and beta, (α, β), Ångström turbidity coefficients
  • the aerosol single scattering albedo, $\omega_a$ (commonly named SSA)
  • the aerosol parameter of asymmetry, $g_a$

and for the absorption of atmospheric gases:
  • the total column ozone content $L_{O3}$ (in Dobson units, DU)

  • the content of precipitable water vapour U (in cm or cm-pr)

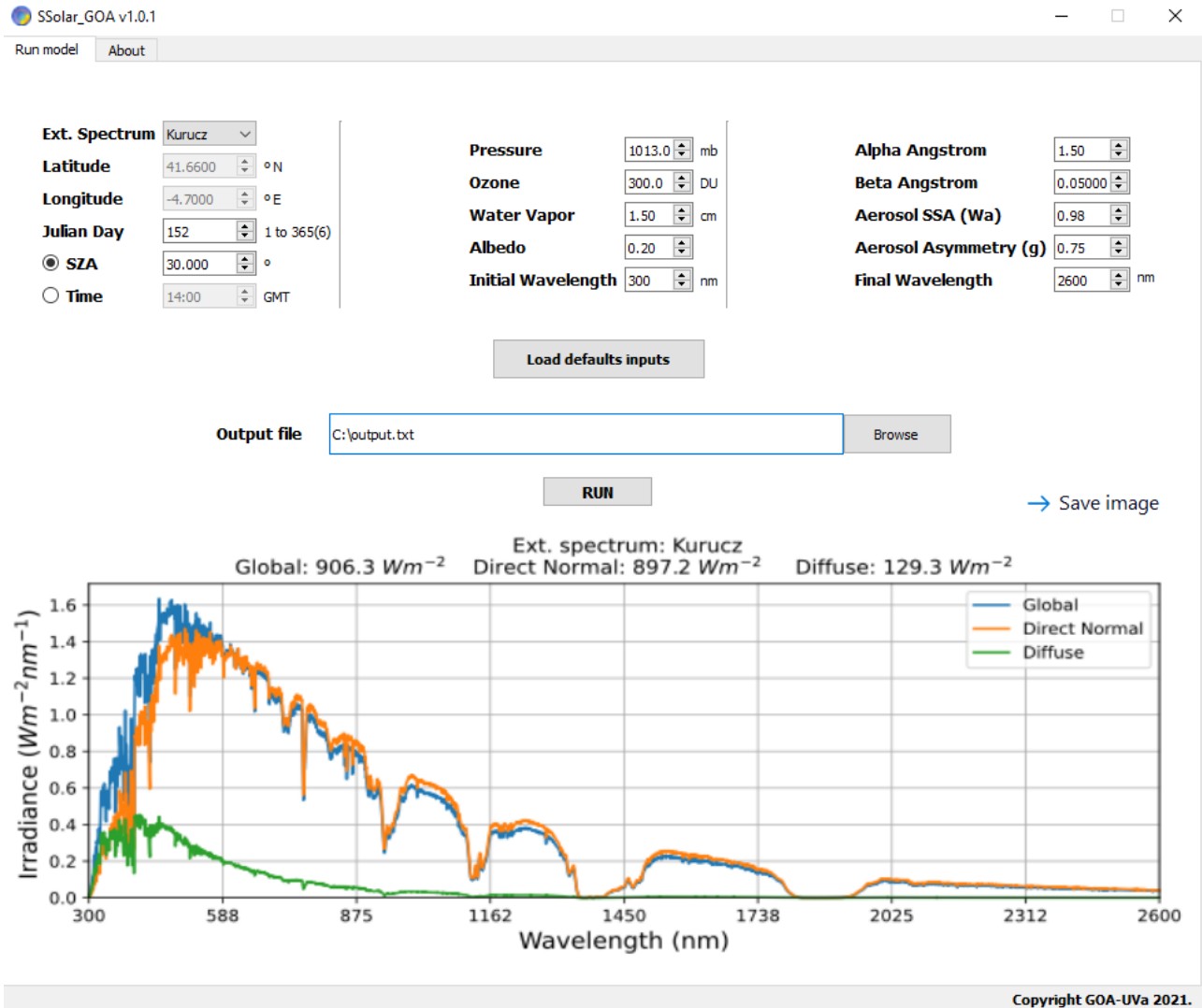

Figure 1: Screenshot of SSolar-GOA v1.0.1 with default inputs configured.

Hence, a total of 10 input parameters are required. N is the Julian day (from 1 to 365) which is required as an input to correct the Earth-sun distance, D, which multiplies the extra-terrestrial irradiance spectrum. The pressure, P, at the site (altitude of the bottom surface) is required to account for the correction of altitude in the Rayleigh scattering optical thickness. The $\alpha$ and $\beta$ Ångström coefficients build the aerosol optical thickness, AOD ($\lambda$), for the whole solar spectral range. Observe that another possible option for the spectral construction of the modelled AOD is to take 4-6 values of the spectral AOD provided by AERONET. As mentioned, the single scattering albedo $\omega_a$ (or SSA) and the asymmetry parameter $g_a$ are taken as constant values because these two parameters are of the second order of importance in relation to the contribution of the aerosol optical

depth. Besides, the Julian day is also required if GMT (UTC) time is used as input instead of the SZA, but in this case it is also necessary to add the latitude and longitude of the site in order to calculate the SZA, and hence a total of 12 parameters must be entered into the model. GMT (or also local time) is currently used when building a set of spectra or when daily solar irradiance values are calculated, since the SSolar-GOA model may also calculate instantaneous integrated irradiance values.

Under clear sky conditions, AOD and water vapour content are the two atmospheric parameters of major importance for irradiance values and ozone and oxygen absorption are also considered because of their strong spectral absorbing features. Other minor absorbing gases, such as $CO_2$ and $NO_2$, are included in the file of absorbing coefficients but are neglected in the running of the current version of the SSolar-GOA model, partially due to their low contribution, but mainly for simplicity and calculation speed.

Generally, the spectral resolution of the model is given by the spectral resolution taken for the spectrum of the extra-terrestrial solar irradiance according to that of the absorption coefficients of atmospheric gases. In our model, we can select three different extraterrestrial work files, given by Wehrli, (1985), Kurucz, (1992) and Gueymard, (2004), as it appears in Figure 1 together with the default input parameters above described.

## 4 Results: performances/validation

### 4.1 Comparison between SSolar-GOA model and libRadtran

The comparison between SSolar-GOA model and libRadtran is carried out as a theoretical exercise, given the latter as a framework reference. For the comparison with experimental spectral irradiance data, the SSolar-GOA model is fed with measured values of the required atmospheric input parameters. Figure 2 shows a set of simulated solar irradiance spectra at sea level by the SSolar-GOA model at three SZAs: 6°, 30°, and 60°, with typical values of the input parameters (given at the

440 top of the figure) under clear sky conditions. This figure illustrates the main characteristics of solar radiation components: horizontal irradiances of direct, diffuse, and global components and the direct normal irradiance. Irradiance values of direct-normal and global solar components show the well-known spectral distribution of solar radiation and their behaviour on the wavelength due to the absorption by atmospheric gases. They increase quickly from near zero at 300 nm to the maximum at visible wavelengths around 500 nm (reaching ~1800, ~1600, and ~800 $Wm^{-2} nm^{-1}$ at SZA= 6°, 30°, and 60°, respectively, for

the global irradiance) and decreasing very slowly along the wavelengths of infrared range. Moreover, the features of water vapour and oxygen band absorptions are the most evident. Overall, the prevalence of global irradiance can be highlighted for low SZA values, but the inverse situation happens when the SZA increases. In this case, direct normal irradiance prevails, starting in the infrared wavelength region and then spreading throughout the whole spectral range, with a greater separation of the spectra of both components.

As typical characteristics, the direct normal irradiance shows a less pointed shape than the global component and a smoother curvature at peak values (from 470 to 700 nm) for increasing SZAs and also the strong variation of these solar irradiances with the SZA in the first part of the spectrum (440 nm to1100 nm) in relation to this last part of the spectrum (1000-2500 nm). On

the other hand, the low values of the diffuse irradiance in relation to the other components under clear skies present the particularity of their minor variations with the SZA and their maximum at the UV region. Thinking about solar radiation as an energy source, it must be noted that irradiance values for SZA=6° are only frequent in sites near the tropics where these low SZAs are reached, while SZAs from 30° to 60° are most frequent in mid and high latitudes where the influence of cos (SZA) on direct horizontal irradiance values are very important, and hence greatly influence the global component.

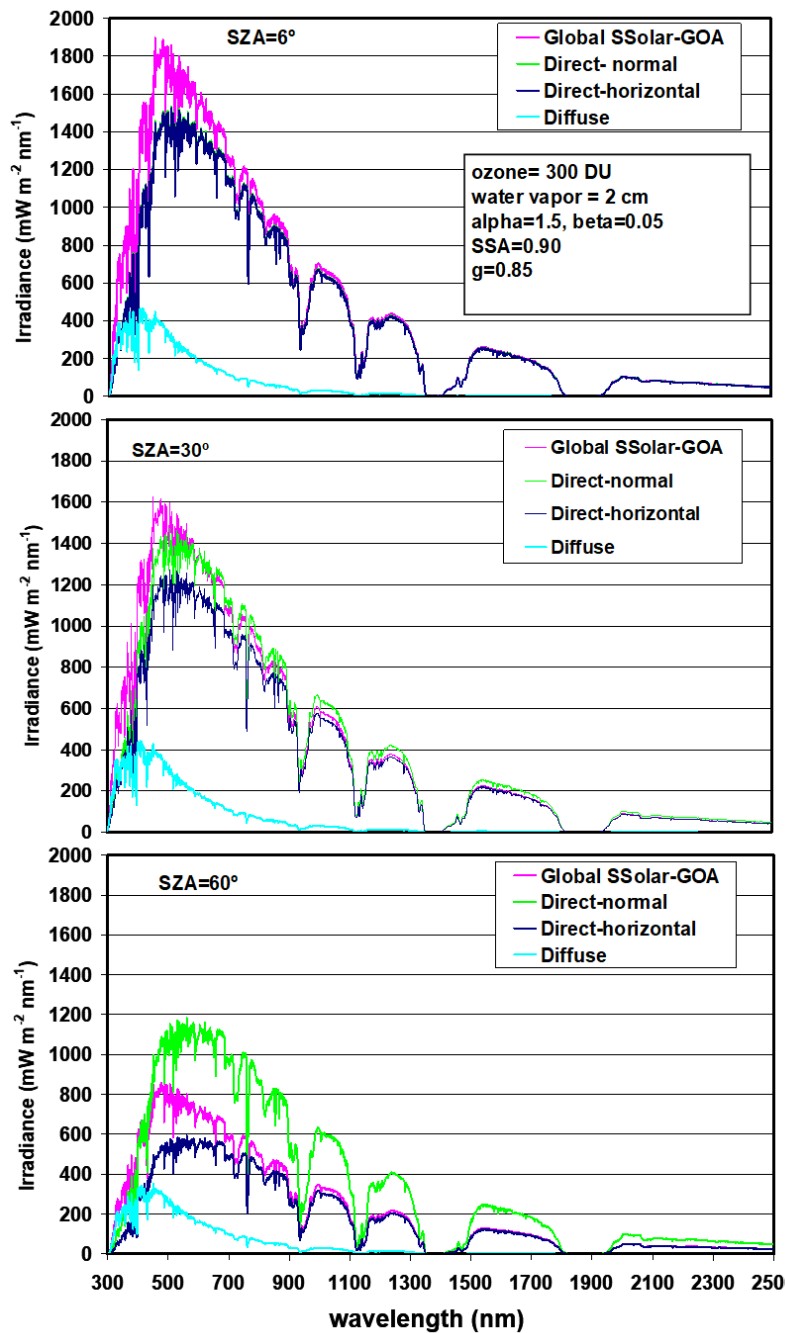

**Figure 2: Global, direct normal, direct-horizontal, and diffuse spectral solar irradiances simulated at sea surface level according to the input parameters shown in the figure for SZAs of 6°, 30°, 60°, respectively (from top to bottom).**

Figure 3 shows the comparison between both models for the direct normal irradiance at SZA= 6°, 30°, and 60° from top to bottom, respectively, with typical values of the input parameters (shown at the top of the figure) corresponding to middle

latitude sites. For better visualization, we have selected the 300-1100 nm spectral range. As can be seen, the results of both models are nearly identical, with relative differences (libRadtran SSolar-GOA/libRadtran) around 0.5% or less than 1% in the non-band absorption regions throughout the entire solar spectral range and covering this large range of SZAs. However, high relative differences with strong and rapid variations, going from positive to negative values (about ±30%), are shown in the regions of the absorption bands of water vapour and oxygen (mainly at 940 nm for water vapour and the oxygen A-band (759-771 nm)). This behaviour must be due to the different spectral resolution in this regions of strong absorption, where the cause could be the slightly different values of the absorption coefficients of each model. Minor differences are found in the visible region due to the smooth and low-absorption of the ozone absorption band (400-650 nm) and due to the very low-absorption of the water vapour bands.

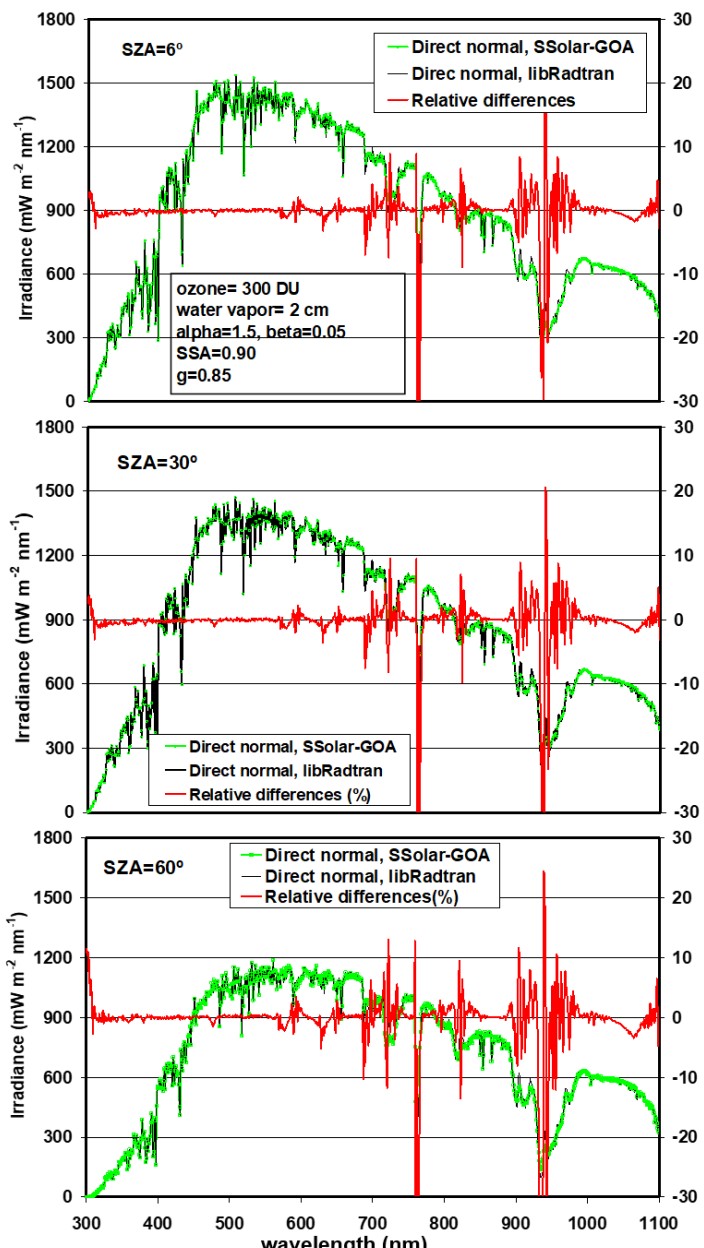

**Figure 3: Comparison between libRadtran and SSolar-GOA models for direct normal irradiance at SZA= 6°, 30°, and 60° respectively (from top to bottom), with input parameters shown at the top of the figure. Right Y axis indicates the relative differences in % in all figures (libRadtran minus SSolar-GOA/libRadtran).**

As already indicated, although both models employed the LOWTRAN7 band model parameterization and similar original coefficients for absorption in the infrared, it seems that the absorption coefficients have undergone a slightly different mathematical handling related to the convolution and interpolation processes. The original LOWTRAN7 absorption

coefficients are given in wavenumber (in unit of cm$^{-1}$) and not in wavelength (in nm). Therefore, the transformation from "cm$^{-1}$" to "nm" gives rise to an inhomogeneous spectral interval of the model, requiring a subsequent interpolation and smoothing (or convolution with a given FWHM) to have a constant step interval, which also depends on the spectral resolution chosen for the model. For example, at the wavelength λ=1 μm (1000 nm), a spectral resolution of 20 cm$^{-1}$ corresponds to 2 nm, but at λ=0.5 μm (500 nm) the spectral resolution is 0.5 nm. Observing the irradiance values of the spectra and the shape-features of these two absorption bands for both models, it is evident that libRadtran presents a slightly higher spectral resolution than the SSolar-GOA model in the regions of gas absorption. For both models, the extraterrestrial spectrum (Kurucz, 1992) is taken with a spectral resolution of 1 nm. Therefore, in the intervals of non-absorption both models present the same spectral resolution and hence they show an exact coincidence for each nm, since the transmittance of scattering processes have a smooth behaviour, and hence the spectral resolution is given by the extraterrestrial spectrum.

For a better visualization of these differences and to confirm the above reasoning, Figure 4 shows in detail the comparison of both models in the region of the 940 nm water vapour absorption band for direct normal (top) and global irradiances (bottom) at SZA=30°. A perfect spectral correspondence (point to point) can be seen in the region of 840-890 nm (just before the "940 nm absorption band" begins) due to the absorption coefficients are zero. On the other hand, one can observe the lower spectral resolution of the SSolar model with a slight smoother behaviour than libRadtran into the "940 nm absorption band". This is because the absorption coefficients of SSolar-GOA model were convoluted with a slit function of FWHM equal to 7 nm.

Although the relative differences in the regions of high absorption by water vapour and oxygen may seem very high, this is the typical behaviour when the spectral resolution of two models is not the same. This is also evident when observing the sharp shape of the A-band of oxygen in the libRadtran package with respect to SSolar-GOA model. Other minor differences between both models are due to the fact that by default the libRadtran considers the complete list of absorption atmospheric gases (such as $NO_2$, $CO_2$, minor gases, etc.) and SSolar-GOA only considers ozone, water vapour and oxygen.

Figure 5 shows the comparison for the global spectral component with the same input parameters and SZAs as Figure 3. The global irradiance differences at SZA= 30° show a slight increase of 1-2% compared with the 0.5% given by the direct-normal irradiance from 330 nm to nearly 700 nm and decreasing for longer wavelengths. The relative differences also decrease with the SZA, and at SZA= 60° the relative differences are less than 1% which is lower than those at 30° and 6°. Relative differences in the regions of the water vapour and oxygen absorption bands show the same variations or features as before. However, a different behaviour in the region of UV ozone absorption band, between 300-320 nm, is observed. The increasing differences range from 0.5% at 330 nm to -25% at 300 nm for SZAs of 6° and 30°, but this trend decreases to 10% for SZA=60°. This feature does not appear in normal direct irradiance values where a very good agreement was observed and only the differences very close to 300 nm increases slightly to 10% for SZA=60° (Figure 3). This problem in the ozone UV absorption band (around 300 nm) will be discussed later.

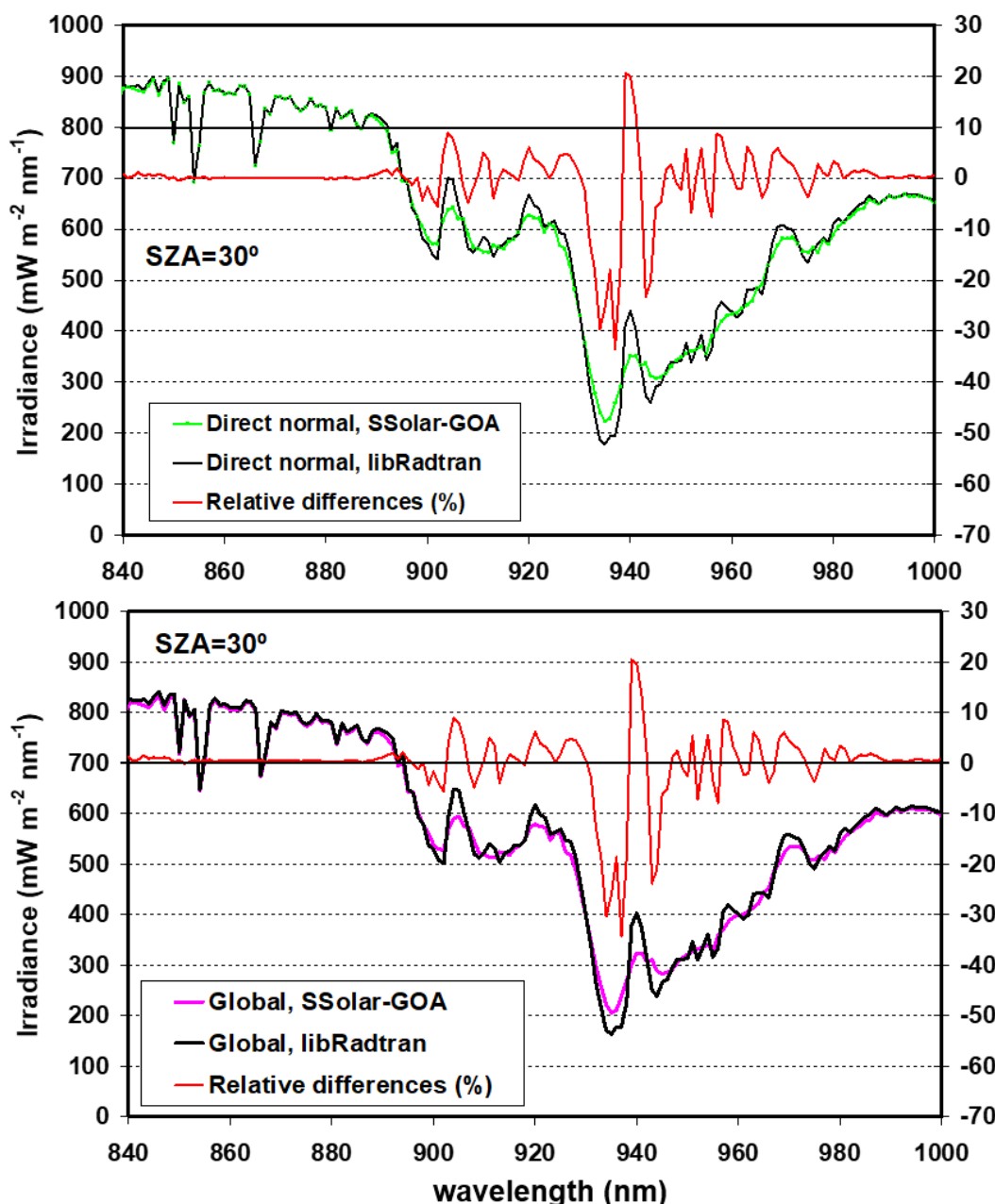

**Figure 4: Comparison between libRadtran and SSolar-GOA models in the region of the 940 nm absorption band of water vapour for direct normal (top) and global (bottom) solar irradiances at SZA=30°.**

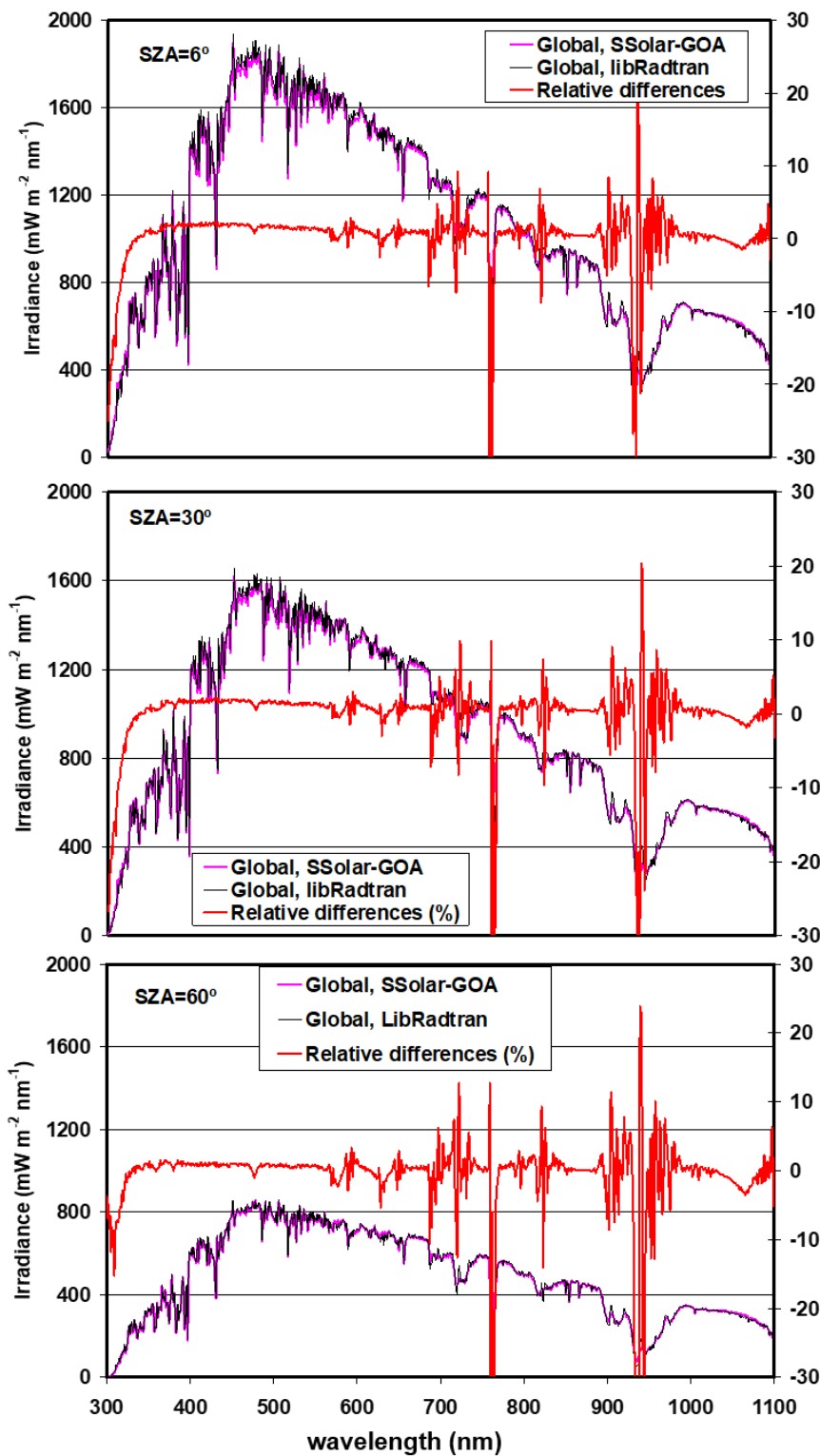

**Figure 5: Comparison between libRadtran and SSolar-GOA models for global irradiance at SZA= 6°, 30° and 60°, respectively (from top to bottom), with input parameters shown at the top of Figure 3. Right Y axis indicates the relative differences in % (libRadtran minus SSolar-GOA/libRadtran).**

Like Figures 3 and 5, Figure 6 shows the comparison for the spectral diffuse component. As mentioned, the diffuse component is obtained as the difference between the global and direct components according to expression (1). Here, the relative differences in the region of non-gas-absorption also increase to reach as maximum 9% in the visible and near infrared range, but this behaviour also decreases at longer wavelengths and with increasing SZA values, with the relative differences ranging from 0-2% at SZA=60°. In general, our model underestimates the diffuse irradiance values for low SZA values in comparison with libRadtran, but there is a good correspondence for the SZA between 40°-60°, and thus a better agreement for mid to low latitudes where these angles are most frequent. The differences are reasonable due to the low diffuse irradiance values under clear sky conditions which accentuates the relative differences. Moreover, it should be emphasised the different physical approaches used by each model for the determination of the diffuse component. LibRadran directly obtains the diffuse irradiance by solving the RT Equation (DISORT solver) while in our model the diffuse component is obtained via the difference between the global and direct-horizontal irradiances.

The problem of the absorption for global irradiance and consequently for diffuse irradiance in the SSolar model in the ozone UV Huggins band may be due to the different treatment of the interaction between scattering and gas absorption. Apart of the above-mentioned procedures for solving the scattering problem (discrete-ordinate/Ambartsumian) libRadtran performed an adequate treatment of the absorption-scattering interaction for the diffuse component (see libRadtran user´s guide, 2015 and 2020), while our model only performs a simple multiplication of absorption and scattering transmittances. Furthermore, the multilayer approach may have also played an important role in this case. Other possible factors, such as the influence of temperature on the ozone absorption coefficients, seem to have had a minor impact because the direct-normal irradiance does not show these high relative differences. On the other hand, this problem does not appear in the absorption bands of other atmospheric gases or in the Chappuis band of ozone because of the lesser absorption of these bands and their rapid saturation compared to the strong absorption of ozone in the Huggins band. However, considering the strong fall that UV irradiances present close to 300 nm (over three orders of magnitude) and the low irradiance values, the increase to -30% of the relative differences (the SSolar-model overestimates UV values) is not so big and in part is enhanced with the artifice due to the different spectral resolution of the absorption coefficients of the two models (this always happens in the regions of strong absorption, as observed).

These Figures are only a visual snapshot of the extensive comparison between SSolar-GOA and libRadtran where hundreds of spectra were compared covering a wide range of SZA values and under the varied atmospheric conditions.

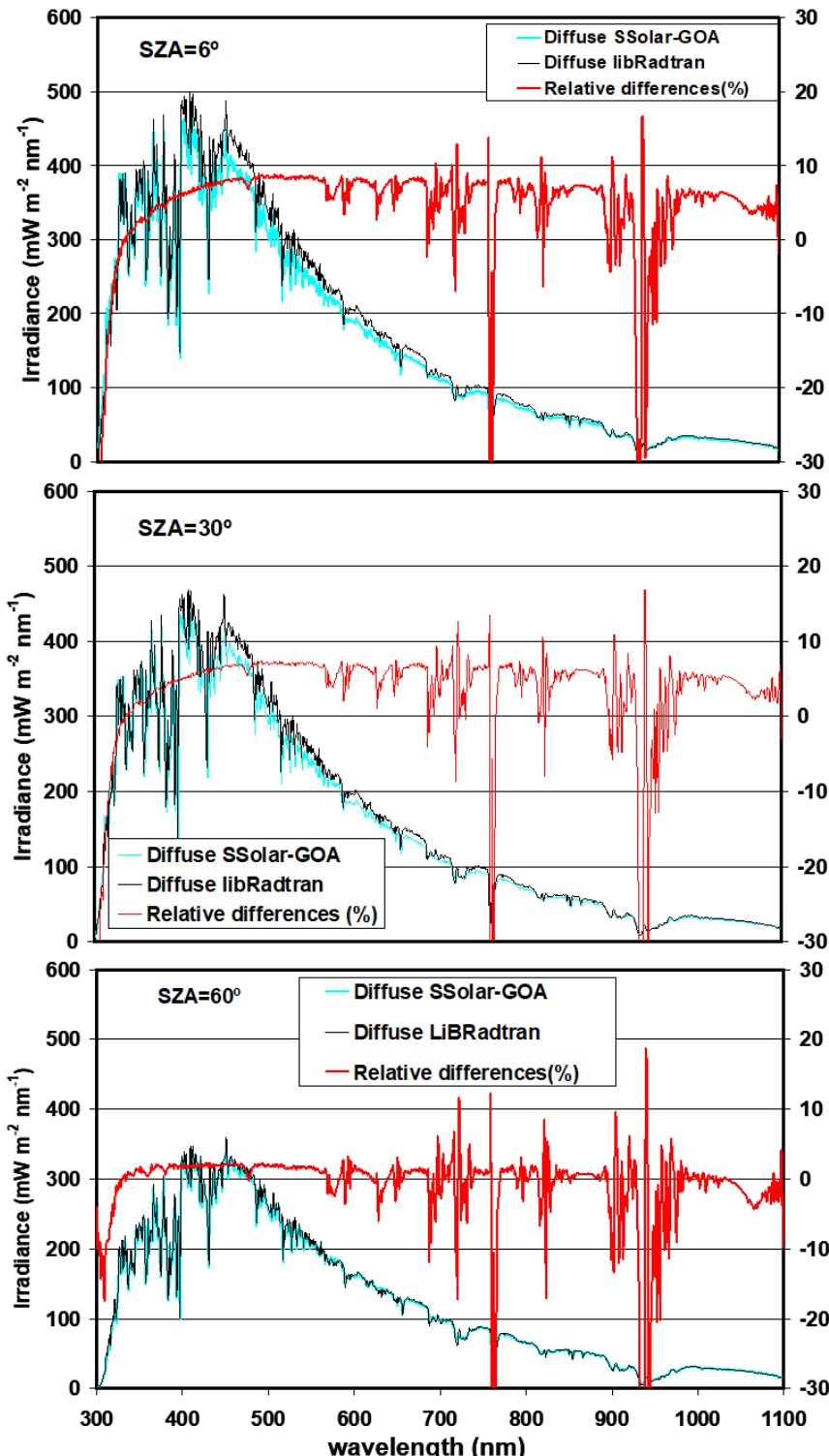

**Figure 6: Comparison between libRadtran and SSolar-GOA models for diffuse irradiance at SZA= 6°, 30° and 60°, respectively (from top to bottom), with input parameters shown at the top of Figure 3. Right Y axis indicates the relative differences in % (libRadtran minus SSolar-GOA).**

A more quantitative evaluation of this comparison was carried out applying linear regression and the RMSE% (Root-Mean-Square-Error, in percentage) statistical indicator using the earlier solar irradiance values and relative differences. The SSolar-GOA model takes the interval 300 nm to 2600 nm as the entire solar range, so this interval was used to apply the linear regression in the comparison between libRadtran and SSolar-GOA for the three SZAs of the earlier figures, as can be seen in Table I. Besides, Table II collects the values of the RMSE% applied to different spectral ranges: UV(300-400 nm), VIS(401-700 nm), NIR(701-1100nm) and the full shown range (300-1100 nm). The reason for analysing this four spectral ranges is due to their different behaviour in the comparison and the fact of not taking the entire range 300-2600 is due to the number of zero values (infinite relative differences) of the irradiance in the water vapour bands existing in the last part of the solar range.

Table I shows the slope (a), intercept (b) and correlation coefficient ($r^2$) for the three SZAs and the three components of solar radiation (it was also added the results for the comparison of measured-modelled data evaluated in the next section). The values of $r^2$ are always higher than 0.99, the slope varies from 1.01 to 1.06 and the intercept from 0.0005 to 0.013 W m$^{-2}$ nm$^{-1}$ (or 0.5 to 13 mW m$^{-2}$ nm$^{-1}$) for the three SZAs, thus resulting in general a very good agreement. Direct normal and global components have similar slopes, near 1, for the three SZAs but diffuse component have a worse value (a=1.06) for SZA of 6º and 30º but improve to 60º with a slope of 1.01, a value similar to the other two components. Intercept (b) values of each case of Table I are very low and they reflects for a given spectrum of a solar component the constant value that SSolar-GOA model always underestimate the irradiances in relation to libRadtran, but this value of intercept refers to all set of wavelengths of the whole spectrum. The equivalent information refers to the slope, therefore linear regression is only a relative good method to know the agreement between the SSolar and libRadtran and more when the $r^2$ reach high values near 1.

Table II gives the values of RMSE% for the three solar components evaluated for the three SZAs and the four intervals in the solar range (for consistency, it was also added the results for the comparison of measured-modelled data which will be analysed in the next section). For the visualized range 300-1100 nm and considering the three SZAs the values are very similar for direct and global components, varying from 5% to 8%, but diffuse component is more stable for the three angles about 9-10%. Very different values are observed for the other intervals with a clear behaviour for lower (3º and 30º) or greater SZA (60º). The VIS range stands out for its low values and low variation of the RMSE% (0.8-1.6%) for direct and global component for the three SZAs, increasing for the diffuse component to about 6-8% for the lower SZA but also decreasing to 2% for SZA=60º. UV interval presents from higher to lower RMSE% for increasing SZAs, from 6º to 60º, with very different values between the three components, very low values for direct component (0.8% to 2.5%), high values for the diffuse (11 to 4.5%) and intermediate values for the global component (6.8%-3.6%), with a substantial improvement in the comparison for SZA around 60º for global and diffuse components. Finally, NIR=701-1100 nm interval shows in general the higher values of RMSE% that the other intervals, in this case always increasing with the SZA for the three components. Global and direct components present similar values (from 6% to 11%) and the diffuse component a less variation, from 9.7% to 13%. Summarising these results,

global and direct components present near similar numbers for the RMSE% for the four intervals and a similar behaviour with the SZA but for the UV interval direct normal component present significant lower values. Diffuse component present the highest RMSE% values, but with a substantial improvement for high SZA, mainly in the UV y VIS ranges. In spite of the

595 valuable information provided by the parameters of the linear regression and the RMSE%, they do not inform in detail which wavelengths fail in the compared irradiance spectrum. Therefore, relative differences evaluated across the spectrum together with the evaluation of a large number of spectra are necessary in order to improve the estimated irradiances of the SSolar-GOA model.

Table I. Linear regression parameters (a is the slope, b the intercept and $r^2$ the coefficient of determination) applied to the irradiance values of the comparison between SSolar-GOA and libRadtran models for the three solar components and the three SZAs. The same for the day 16 and 19 of figures 8 and 9 of the comparison between measured LI-1800 and modelled SSolar-GOA irradiance spectra.

| Linear regression | GLOBAL | | | DIRECT Normal | | | DIFFUSE | | |
|---|---|---|---|---|---|---|---|---|---|
| | a | b | $r^2$ | a | B | $r^2$ | a | b | $r^2$ |
| SZA=6° | 1.02 | 10 | 0.99 | 1.01 | 10 | 0.99 | 1.06 | 0.0 | 0.99 |
| SZA=30° | 1.02 | 10 | 0.99 | 1.01 | 10 | 0.99 | 1.06 | 0.4 | 0.99 |
| SZA=60° | 1.01 | 7 | 0.99 | 1.02 | 13 | 0.99 | 1.01 | 0.5 | 0.99 |
| Day 16 | 0.98 | 10 | 0.99 | 1.01 | 10 | 0.99 | 0.98 | 9 | 0.94 |
| Day 19 | 1.04 | 9 | 0.99 | 1.0 | 7 | 0.99 | 1.21 | 10 | 0.99 |

Table II. RMSE in percentage (%) evaluated in the comparison between SSolar-GOA and libRadtran models for the three solar components
and the three SZAs. The same for the day 16 and 19 of figures 8 and 9 of the comparison between measured LI-1800 and modelled SSolar-GOA irradiance spectra.

| RMSE % | GLOBAL | | | | DIRECT Normal | | | | DIFFUSE | | | |
|---|---|---|---|---|---|---|---|---|---|---|---|---|
| | UV | VIS | NIR | FULL | UV | VIS | NIR | FULL | UV | VIS | NIR | FULL |
| SZA=6° | 6.3 | 1.6 | 6.6 | 5.3 | 0.8 | 0.8 | 6.6 | 4.6 | 11. | 7.6 | 9.7 | 9.3 |
| SZA=30° | 6.8 | 1.5 | 7.4 | 5.8 | 0.9 | 0.9 | 7.2 | 5.1 | 11. | 6.3 | 10.0 | 8.9 |
| SZA=60° | 3.6 | 1.4 | 11.2 | 8.0 | 2.5 | 1.3 | 11 | 7.9 | 4.5 | 2.0 | 13.3 | 9.6 |
| Day 16 | 12 | 1.5 | 4.7 | 5.7 | 9.9 | 1.6 | 5.0 | 5.1 | 16 | 16 | 34 | 26 |
| Day 19 | 18 | 5.4 | 5.7 | 8.3 | 11 | 1.6 | 5.0 | 5.6 | 37 | 26 | 66 | 51 |

## 4.2 Comparison between SSolar-GOA and spectral solar irradiance measurements

To validate the SSolar-GOA model, we selected specific well-suited spectra from our irradiance solar databank. Thousands of solar irradiance spectra have been measured over the past 25 years by GOA for different research activities, most of them focused on atmospheric studies for the determination of atmospheric components (Cachorro et al., 1987b, 1996, 1998, 2000b, a; Vergaz et al., 2005) and modelling of solar spectral radiation. In Cachorro et al. (1985, 1987a, c), one of the first comparisons between field experimental spectral solar irradiance measurements and their modelling with simple spectral solar radiation models can be seen. Detailed radiative transfer models have been also used and compared with experimental spectral solar irradiance data (Cachorro et al., 1997; Durán, 1997; Utrillas et al., 2000; García et al., 2016).4.2.1 LI-1800 measured spectra. A comparison of the SSolar-GOA model and field measurements with the LI-1800 spectroradiometer was carried out as already mentioned. Figure 7 shows 26 spectra of global solar irradiance measured throughout the day of 16 July during the Veleta campaign (Estellés et al., 2006; Alados-Arboledas et al., 2008). This campaign was carried out in July 2002 with the aim of aerosol characterization, making an extensive comparison of aerosol properties retrieved by different instruments, mainly Cimel sun-photometers and LI-1800 spectroradiometers. The campaign was carried out at several locations in the Granada province (Andalusia region, Southern Spain). Specifically, the comparison illustrated in Figures 7 and 8 corresponds to the rural village of Pitres, (1300 a.s.l.) in the Alpujarras region, an area at the southern slope of the Sierra Nevada Mountain Range.

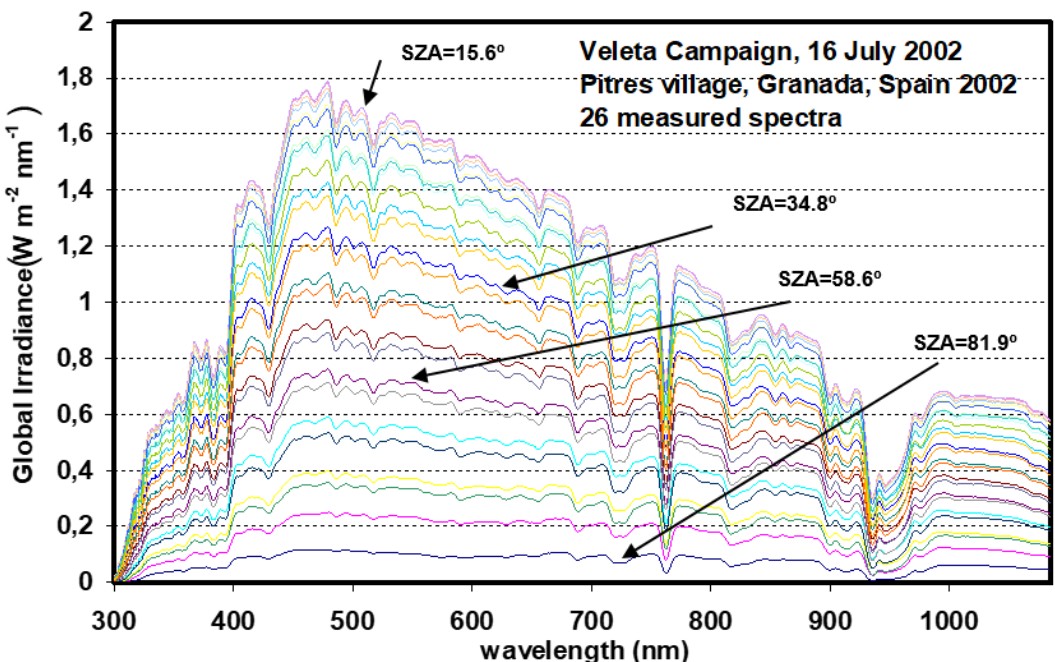

**Figure 7: Solar global irradiance spectra measured by LI-1800 during the afternoon of 19 July 2002, during the Veleta campaign at Pitres (Granada, Spain).**

The validation process requires accurate input model parameters not always available, but in our case, they were provided by various Cimel sun-photometers installed for the Veleta-2002 campaign (as explained in Estellés et al. (2006) and Alados-Arboledas et al. (2008)). The water vapour content was provided by one of these Cimel sun photometers connected to AERONET. The ozone vertical content was obtained by the daily values provided by the TOMS satellite sensor. Due to the error associated with the determination of the β turbidity parameter and the fact that AERONET did not provided it, the value

of this parameter was replaced by the aerosol optical depth at 1020 nm. Since Cimel and LI-1800 measurements are not exactly coincident in time, the closer measured values were taken or the interpolated data in between. The aerosol single scattering albedo ($\omega_a$) and the asymmetry parameter ($g_a$) were taken as constant with the wavelength, as a first simple approach to the modelling as explained above. The value of $g_a$ was taken as an average of 0.65 for that day, as was $\omega_a$, which had a value of 0.99 (both values were provided by AERONET). These two values are reasonable for non-absorbing aerosols, such as those

characteristics of clean rural areas such as Pitres. All these values of the input model parameters appear in Figure 8 in order to model the three components of solar radiation.

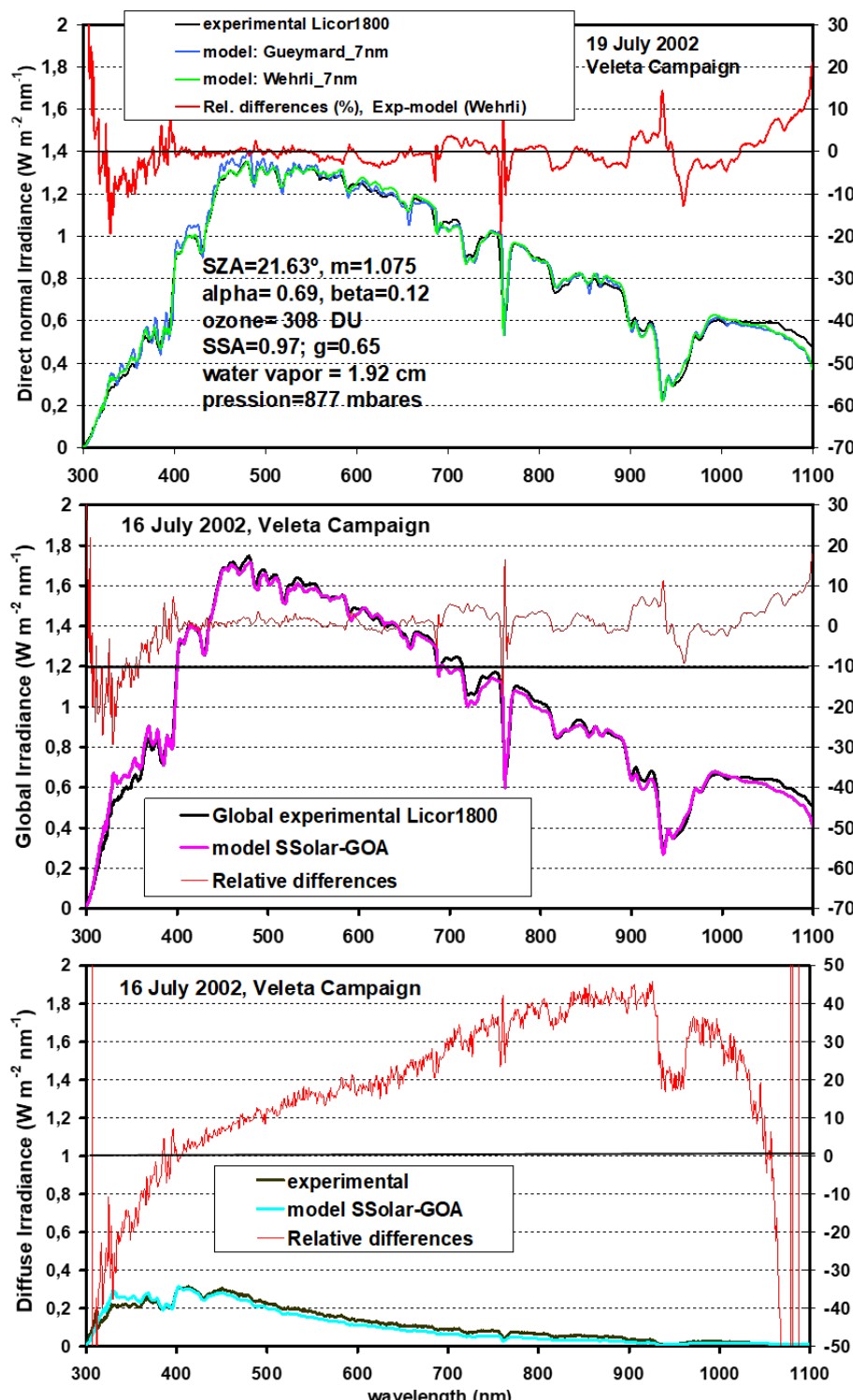

**Figure 8: Comparison between the LI-1800 measurements and the SSolar-GOA model for direct, global, and diffuse spectral irradiances (from top to bottom) for 16 July 2002, at Pitres (Granada, Spain). The input parameters are specified at the top. Right Y axis indicates the relative differences (%, in red colour) of measured minus modelling data (only for the output data taken from the Wehrli spectrum).**

Figure 8, corresponds to 16 July of Veleta Campaign, where the direct normal and global horizontal components were measured 2 minutes apart: at 11.28 GMT time for direct normal component and 11.30 GMT for the global component, with a nominal SZA of 19.15 and 19.45, respectively (the air mass values were 1.057 and 1.061, respectively). Bear in mind that the LI-1800 takes about 40 seconds to measure a spectrum from 300-1100 nm, and we assign a unique time-value for the measured spectrum. Therefore, the time difference between direct and global spectra is non-significant in terms of modelling. An excellent agreement is obtained between measured-modelled data for direct normal irradiance values, with relative differences ranging from 1 % to 3% in the visible range (400-700 nm) and less than 10% in other spectral ranges. RMSE% of Table II gives a 5.1% for the 300-1100 nm, 1.6% for the VIS and 9.9% for the UV and Table I gives 1.01 for the slope, 11 mWm$^{-2}$ nm$^{-1}$ for the intercept and 0.99 for r$^2$. We have used the spectrum of Wehrli (Wehrli, 1985) convoluted with the spectroradiometer slit function represented by a triangular function of 7 nm FWHM since the original file has a spectral resolution of 1 nm. We call attention to the observed lesser differences of the already mentioned oxygen and water vapour bands because of the similar spectral resolution between our model and the measured data from the LI-1800 in relation to the above comparison with the libRadtran. The observed differences around 1100 nm are due to a specific problem of heating in the LI-1800 spectroradiometer (bear in mind that this instrument is not thermally stabilized). The temperature in Pitres in July reached up to 35 °C, but this problem disappears for lower temperatures.

Similar results were obtained in the comparison of the global irradiance spectrum in this case and whose statistical indicators are listed in the table I and II. Diffuse irradiances show greater disagreement with underestimated values for the infrared and overestimated values for UV and good agreement around 400 nm. The differences range from -40% to 40%, the slope of the linear correlation is 0.98 and the r$^2$ falls to 0.94. The RMSE% for the whole measured spectral range is 5.7% with low values in the VIS and higher in the UV and NIR as can be seen in Table II. However, considering the low values of diffuse irradiances for clear skies and the associated uncertainty, it can be said that these differences are into a reasonable concordance. As already mentioned, both the measured and the modelled diffuse irradiance values were obtained as the difference between the global irradiance and the horizontal direct irradiance, where uncertainties are added. The approach of assuming an isotropic model to evaluate the horizontal direct spectral irradiance (bear in mind the factor giving by cos(SZA)) entails a high uncertainty that is difficult to assess, and that is more pronounced considering that Pitres is on the slope of Sierra Nevada.

Figure 9 corresponds to 19 July where direct normal and global components were measured 2 minutes apart: at 13.26 GMT time for direct normal component and at 13.28 GMT for the global component, with a nominal SZA of 21.63 and 21.92, respectively (the air mass values were 1.075 and 1.077, respectively). The value of alpha=0.69 and SSA=0.97 parameters corresponded to a desert dust aerosol type, since a low-moderate intrusion of desert-dust arrived to this area on 19 July. Table I reports 1.0 for the slope, 7 mWm$^{-2}$ nm$^{-1}$ for the intercept and 0.99 for r$^2$ and Table II gives a RMSE% of 5.6% for the 300-1100 nm, and 11%, 1.6%, 5% for the UV, VIS and NIR spectral ranges respectively. Therefore, the modelled direct normal

irradiance shows a very good agreement with the measured data as shown earlier in Figure 8, but in this Figure 9 we have also added the simulated output irradiance taken the Gueymard (Gueymard, 2004) extraterrestrial spectrum (it was also convoluted

as before the Wehrli (1985) spectrum). In this case, it can be observed some slight differences between experimental and modelled irradiances between 400 and 500 nm.

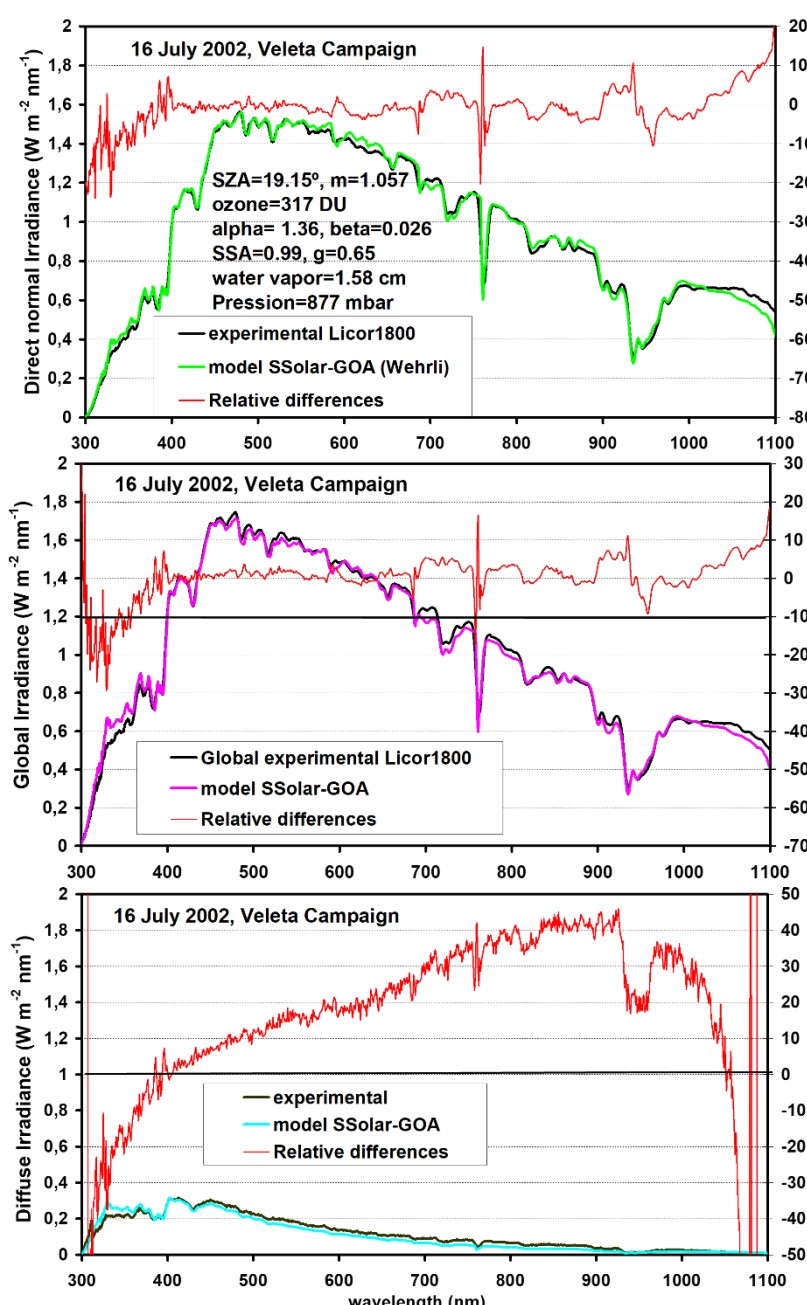

**Figure 9: Comparison between the LI-1800 measurements and the SSolar-GOA model for direct, global, and diffuse spectral irradiances (from top to bottom) for 19 July 2002 at Pitres (Granada, Spain). The input parameters are specified at the top. Right Y axis indicates the relative differences (%, in red colour) of measured minus modelling spectra (only for the output data taken from the Wehrli spectrum).**

These differences are due to the differences in the original extraterrestrial spectra, as can be seen in Figure 20, where both spectra are compared and where the well-known Kurucz extra-terrestrial spectrum was also added to strengthen the comparison. The differences between Wehrli and Gueymard spectra in terms of quantity are around ±5% as maximum, due to the spectral variability in the UV-Visible region (300-500nm) if compared to the smoother behaviour in the infrared. However, both spectra present greater relative differences with that of Kurucz, with positive and negative values that reach a maximum of 10-15% in the mentioned 300-500 nm region. Therefore, it is important to note the observed differences between the solar models and spectral measurements due to the uncertainty associated with the different extraterrestrial spectra.

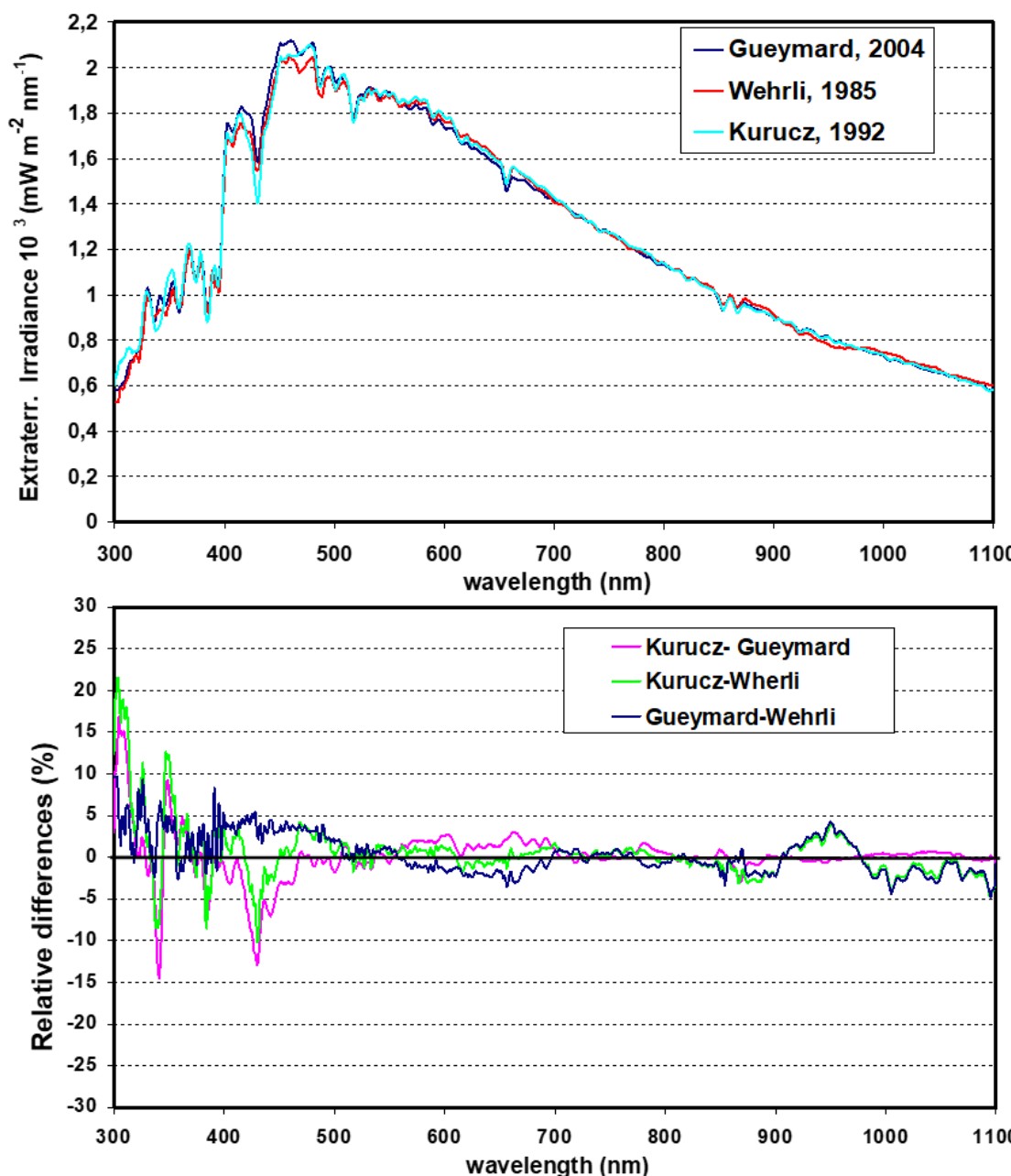

**Figure 10: Comparison between the extraterrestrial solar irradiance spectra given by Gueymard, Wehrli, and Kurucz (see references) convoluted with a spectral triangular slit function of FWHM of 11 nm.**

Modelled global irradiance for 19 July shows greater values than measured ones with differences around 5% in the visible region which are greater than those on 16 July. As expected, diffuse irradiances also show important differences with a higher
overestimation of modelled data derived from the earlier overestimation of global spectral data. However, we can observe the

different spectral behaviour shown by the relative differences on days 16 and 19. Day 19 presents more stable behaviour with always negative differences ranging from 20% to 40%. Certainly, diffuse modelled data do not present good agreement for low SZA angles, but an improvement is found for higher SZAs (see next section). The RMSE% and parameters of linear regression for these two solar components can be also seen in Table I and II. For global radiation Table I reports 1.04 for the slope, 9 mWm$^{-2}$ nm$^{-1}$ for the intercept and 0.99 for r$^2$ and Table II gives a RMSE% of 58.3% for the 300-1100 nm, 18% for the UV, 5.4% for the VIS and 5.7% for the NIR spectral ranges. For diffuse radiation RMSE% values increase considerably, varying from 37% to 66% depending on the selected spectral range. As mentioned, these low values of diffuse irradiances enhance the percentage quantities. Slope (1.21) get worse but reflecting the overestimation of modelled values, and intercept (10 mWm$^{-2}$ nm$^{-1}$) and determination coefficient (0.99) give good values.

### 4.2.2 ASD-FR-Pro measured spectra

Taking advantage of the high temporal resolution of the ASD spectroradiometer, this instrument was programmed in our field's campaigns in sequences of hours to measure one spectrum (from 350 nm to 2500 nm) every minute. A set of 890 global solar spectra were measured throughout the day of 29 July 2008, at the site of Andenes on Andøya Island in the Verterålen Archipelago in Norway. Because the great number of spectra, we selected different wavelengths and observed their behaviour throughout the day. Figure 11 (top) shows the measured (dark-blue points) and modelled (continuous green line) global irradiance values at the wavelength of 440 nm as a function of GMT time. The values of global irradiance at 440 nm are drawn from each measured spectrum. To generate the modelled values, a constant aerosol optical depth throughout the day of AOD (440 nm) =0.14 was considered, in accordance with the mean value of the day and the behaviour of the aerosol optical depth during the day. Precisely, Figure 12 shows the time evolution of AOD at different wavelengths and the alpha parameter on 29 July measured by the Cimel sun-photometer of the Andenes-AERONET station.

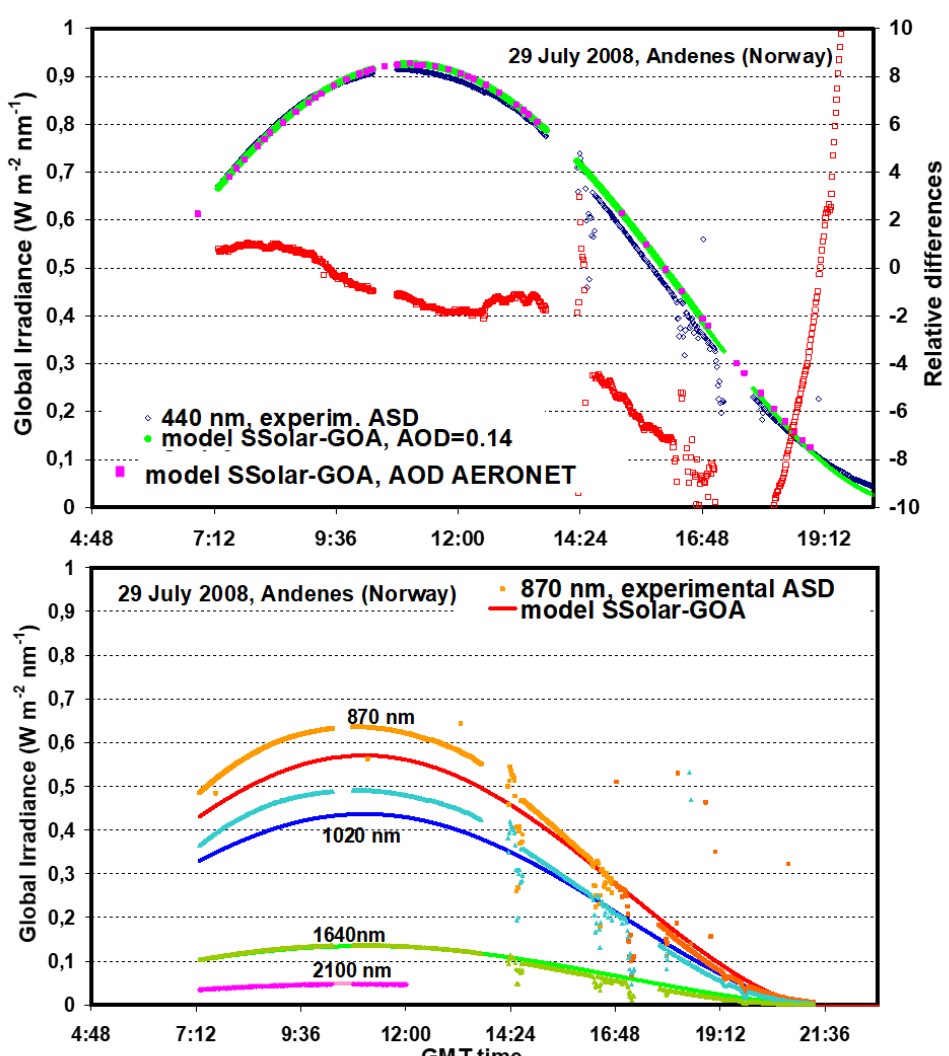

**Figure 11: Comparison between the ASD measurements (dark blue line) and the SSolar-GOA model (green line) for the global spectral irradiance at 400 nm as a function of GMT time on 29 July 2008, at Andenes (Andøya Island, Norway). The rose points overlapping the green line are also modelled points at the specific time, and the AOD values are given by AERONET Cimel sun-photometer. At the bottom graph, the lines also give measured and modelled global irradiance values for different wavelengths (orange- red is 870 nm, light and dark blue is 1020 nm, dark and light green is 1640 nm, dark and light rose is 2100 nm).**

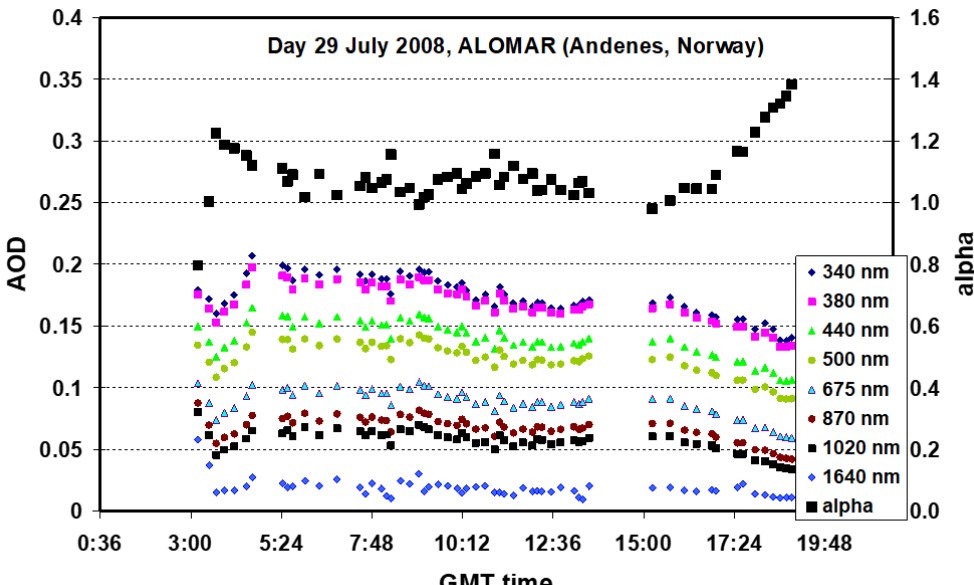

**Figure 12: Time evolution of AOD at different wavelengths and alpha parameter on 29 July 2008 at Andenes (Andøya, Norway).**

Therefore, in order to account for the variability of the AOD during the day, we have taken these values as the input in the model resulting in rose points, just over the green line. In addition to the aerosol parameter provided by AERONET, ozone and water vapour content were also taken from the AOD file of AERONET (level 2, quality assured). The good agreement demonstrates the low variability of AOD throughout the day and the correct approach for a fixed AOD value for modelling

the entire day. A very good agreement is obtained with relative differences (ranging about ±2%) in most of the central hours of the day and falls to -10% thereafter, wherein the SZA reaches values close to 90º and the relative mass reaches the value of 40 (at these points, the relative differences grow rapidly because the very low irradiance values).

The observed scattered points are due to clouds because the measured spectra are not screened. Usually if significant cloudiness was observed, the system was stopped, but often the observed breakdown in the line of global measured values is because the

ASD system was also arranged to measure the zenith radiance. During the day, we alternated some periods to measure the global irradiance and others to measure the zenith radiance, but on day 29 most of the measured values were of global solar irradiance.

At the bottom of Figure 11, a similar graph is shown but for the wavelengths of non-absorption of 800 nm, 1020 nm, 1640 nm, and 2100 nm. For the 800 nm wavelengths, the orange points are the measured values and the red line contains the modelled

values. The same is true for the 1020 nm (light blue points measured and a dark blue modelled line), 1640 nm (dark green points measured and a light green modelled line), and 2100 nm (rose points measured and a light rose modelled line) wavelengths. As stated above, the modelling was carried out with a fixed AOD value at each specific wavelength taken from the AERONET data according to Figure 12. While longer wavelengths of 1640 nm and 2100 nm show a perfect agreement between the measured and modelled values as 400 nm, the other two wavelengths at the near infrared, 870 nm and 1020 nm,

give a greater disagreement of about 10-12% in the interval of time around the central hour of the day and decrease at 16 GMT. For a better visualization of this Figure 11, the values after 12 GMT time at the 2100 nm wavelength have not been drawn but this wavelength also gives a perfect concordance.

These observed differences at these infrared wavelengths may be due to different causes: a) a much greater than usual error due to ASD calibration at these wavelengths; b) for global radiation measurements, special care must be taken with the
horizontal levelling of the cosine receptor sensor, taking into account that this platform is moving for the alternate zenith radiance measurements; c) the error linked to the modelling refers to the complete and perfect curvature of the modelled spectra of solar irradiance which is not easy and even less so if we model a wide spectral range. The curvature of the irradiance spectrum is governed by the shape of the curvature of the AOD, that is, by the dependence of AOD on wavelength. In our modelled values, this curvature is constructed by the pair of values from the Ångström α-β turbidity parameters which only
gives a linear behaviour on the plot of log-AOD versus log-λ, while real aerosol showed an accentuated curvature on this type of plot. Nevertheless, the modelling can be improved by taking two pairs of α-β values applied to different spectral intervals or by taking 5-6 values of measured AOD, but all this entails more complicated input model parameters. For example, the alpha-beta values determined in the visible region are not recommended to be applied in the UV region. It is easy to observe how in our model the UV region presents greater relative differences than other parts of the spectrum -when considering non-
gas absorption regions.

However, more similar measured-model values would be expected in Figure 11, bearing in mind that the modelling at these selected wavelengths is more accurate than the modelling of the entire spectrum because in this case it contains the exact AOD value at these wavelengths. Besides, in the above comparison with the LI-1800, we have also often observed these differences between measured-modelled values for global irradiances of about 10-15%. Therefore, an error in modelling added to
calibration errors can reach these values.

 Figure 13 shows the measured and modelling values of three specific spectra on 29 July, from 350 nm to 2200 nm, at SZAs of 50.86, 67.29 and 82.59, respectively. The three spectra show a slightly different agreement with the modelled data. A notable disagreement is observed between the measured-modelled spectrum at 10.48 GMT (SZA=50.86), with relative differences reaching 10-15%. Spectra at SZAs of 67.29 and 82.59 show a better concordance, with relative differences of about 2-10%.
These are the same results observed in Figure 11 when analysing discrete selected wavelengths throughout the day, but now giving the overall behaviour of the whole spectrum. Certainly, the spectrum at SZA=82.59 (m=7.3) represents an extreme situation with very low spectral irradiance values, which may be of interest for some applications, such as the determination of the amount of absorbing gas.  However, these cases are of little interest in solar energy resource at middle latitudes, but not negligible in very low latitudes since there are a large number of hours with this insolation.

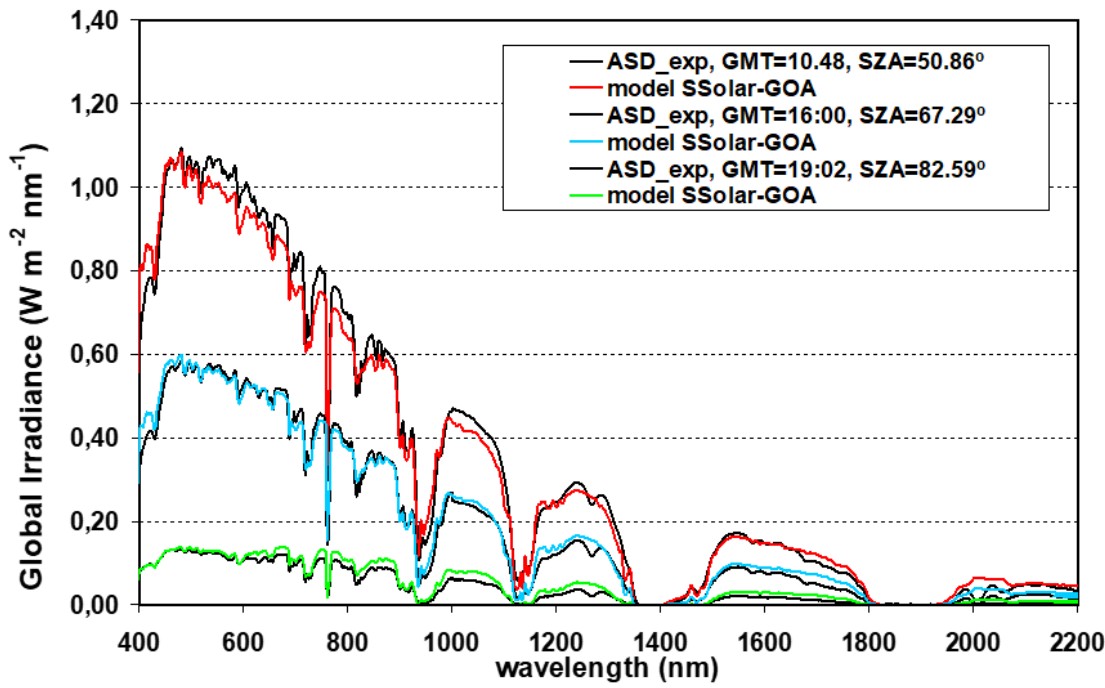

**Figure 13: Comparison between the ASD measured and modelled SSolar-GOA global irradiance spectra covering the spectral range from 350nm to 2200 nm, taken on 29 July 2008 at Andenes (Andøya, Norway) at three SZAs. Input aerosol parameters are obtained from the values shown in Figure 12 (see text).**

## 5 Discussion and conclusions

Despite the abundant bibliography and research about solar radiation models, there exists a broad gap between the different research communities that develop and use/apply solar radiation models (i.e., between the models used by the solar energy community, satellite remote sensing, or in the same climate-atmospheric area). Certainly, each community have their own necessities and objectives and hence they may use solar radiation models for many distinct applications. On the other hand, the number of different methodologies developed to solve the process of scattering and absorption of atmospheric components,

from complicated methods to simple approaches, constitutes a rich and varied field of study. The solar energy community mainly develops and applies solar radiation models based on empirical expressions fitted on measured solar radiation data, while in the climate/atmospheric field a more theoretical-physical foundation is contained in the radiation models. Therefore, this work seeks to decrease this gap so that potential users who are not very familiar with Radiative Transfer Theory can make use of solar physical radiation models if they are presented under simple parameterized expressions, based on a set of input

parameters easy to use and understand.

The evaluation of the diffuse component is generally a more complicated problem, and most of the models are based on the solution of the RTE for the scattering process. However, here RTE solving is replaced by a different methodology developed by Ambartsumian and represented by an uncomplicated analytical function which expresses the transmittance of total

scattering of a mixed molecule-aerosol layer, which is really the core of the model. Although this analytical transmittance is a function of more unknown aerosol parameters, such as the single scattering albedo and the parameter of asymmetry, the aerosol optical depth is the most relevant parameter which drives the model and this is provided in many sites around the world by AERONET network.

The SSolar-GOA model is structured based on a single layer for the entire atmosphere, therefore the evaluation of solar irradiances must be made at the bottom surface but the altitude of this surface is not necessary the ground level, it may be defined by the user (e.g., on top of a mountain, the flight level of an airplane, or the see surface) but taking into account the adequate input parameters. The method of Ambarsumian also evaluated the reflectance of the mixed layer of molecules-aerosols and this new magnitude will be considered in further development of the SSolar-GOA model, extending it to other possible applications, mainly in flight platforms and satellite remote sensing areas.

On the other hand, to take a unique atmospheric layer instead of multiple layers is not a great handicap for the estimation of solar irradiances under clear skies if their evaluation is based on the LBL approach.The main contribution to global solar irradiance at the lower level surface under clear skies is given by the direct component, where its contribution is about 80-62% for the SZA in the range from 20º to 70º under current atmospheric conditions of aerosol load ($\sim$ AOD(500nm)= 0.1) and water vapour content ($\sim$ 1.5 cm), being these two atmospheric components the most influential.

The direct normal spectral component based on the LBL law and expressed as the product of exponential function transmittances results in concise and computationally undemanding formulation. Importantly, it was shown that the assumption of a single layer of aerosol-molecules instead of multiple layer atmosphere does not have significant influence over the calculated values of the spectral solar direct irradiance, thanks to these exponential functions that drive the absorption and scattering processes. The multiplication of exponential function is equivalent to the sum of its exponents and the total optical thickness of the whole atmospheric one layer is the sum of the multiple layers and hence the same value is obtained. Although this fails for gas absorption because the dependence of absorption coefficients on pressure and temperature, the difference on spectral irradiance values are not relevant when we want to estimate solar radiation at ground level, as those measured by our spectroradiometers, or for many applications in solar energy, agriculture and forest, ecology, where an accuracy about 5-10% may be sufficient.

Depending of the required level of accuracy for the solar spectral irradiances SSolar-GOA model can provide them as input variables in other radiative transfer models applied to vegetation studies, as SAIL and PROSAIL models (Jackemoud et al., 2009, Berjón et al., 2013) or as part of sub-models in the new Earth System Models (ESMs), as SCOPE (Yang et al. 2021) or CliMA (Braghiere et al. 2021). Solar radiative transfer models applied to vegetation to retrieve biophysical plant parameters not only share many methods and concepts with RT models developed to atmosphere but they are joined or combined when satellite remote sensing data are acquired for this objective.

Climate models and forecast weather models (Sukhodolov et al., 2014) do not use spectral solar radiation models because they need a rapid evaluation which is covered by the "integrated or broad-band" solar radiation models. Although many of them consider the entire solar spectrum divided in various intervals or spectral bands using the K-correlation method as the most

common to account for the absorption of gases. Therefore, in this areas of application the SSolar-GOA model may be useful as a rapid testing of these "broad-band" models since it also gives as output the integrated values of the irradiances for the three components. The inclusion of an effective plane-parallel cloud layer is also a feasible possibility taken a parameterized cloud scheme (Liou, 1992) which can increase the potential of SSolar-GOA model, but it must be in mind that SSolar-GOA model was designed as a clear sky simple model, easy to use, which cannot to compete with multilayer RT Codes that solve the RT equation. To authors´ knowledge it is not easy to find in the literature a spectral model of similar characteristics, being the most similar the SMART model and in this context see the recent publication of Gueymard (2019) about the variety of applications where this model has been used in the last 20 years.

The performance of the SSolar-GOA model is clearly demonstrated by the comparative task with the libRadtran model, where a very good agreement is obtained. Both are based on a similar evaluation of the direct component thanks to the LBL law. The discrepancies in the diffuse solar spectral component are mainly due to the different theoretical treatments of the interaction scattering-absorption processes between both models. Certainly, the comparison with experimental data does not reach the same level of agreement as before, but it highlights the difficulty of spectral solar radiation measurements. The proposed model has a strong physical base and due to its simplicity, accuracy, and rapid runtime it is well suited to evaluate the three components of the spectral solar radiation data – today required by many different applications – and is therefore open to very different type of users.

***Code and data availability***. The SSolar-GOA model version 1.0 is open-source and can be accessed at a DOI repository: https://doi.org/10.5281/zenodo.5796545 (Cachorro et al., 2021). This code has licence GNU General Public License v2.0 or later. The dependencies and install instructions are in Readme file. For windows users a binary package has been generated which can be download from http://goa.uva.es/ssolar_goa-model/.

***Author contributions***. The model was designed, developed and evaluated by V. Cachorro (with first software versions in FORTRAN). Spectroradiometer calibration and maintenance was performed by A.M. de Frutos and C. Toledano. Measurements were carried out by the different members of GOA-UVa team over the last 25 years. The first current software version of the SSolar-GOA model in Python was built by V. Molina Garcia (currently at DLR, Oberpfaffenhofen, Germany). J.C. Antuña-Sanchez makes the current final software version of the model and the internet platform for users. V. Cachorro wrote the paper and J.C. Antuña-Sanchez prepared the final manuscript for Journal submission. All authors have read and agreed to the published version of the manuscript.

***Competing interests***. The authors declare that there is no conflict of interest.

***Acknowledgements***. The authors gratefully thank AERONET/RIMA for the aerosol products, ALOMAR Laboratory (Andøya Space Centre, Andenes, Norway), and the GOA-UVA team for the spectral solar radiation measurements and all kinds of help.

Special thanks to all people who took part in the "Veleta 2002 campaign". Special thanks to Rosa D. García of the Izaña Atmospheric Research Center (AEMET) for its valuable contribution in the simulations with libRadtran and SSolar-GOA models.

*Funding:* This research was funded by the Spanish "Ministerio de Ciencia, Innovación y Universidades (MICINN)" under Reference Project RTI2018-097864-B-I00, and by Junta de Castilla y León, (reference VA227P20).

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
