# Peer review of "SSolar-GOA v1.0: a simple, fast, and accurate Spectral SOLAR radiative transfer model for clear skies"

_Geoscientific Model Development, 2021_

## Referee Comment (RC1)

**Nina Črnivec's Review of the paper:**

**"SSolar-GOA v1.0: a simple, fast, and accurate Spectral SOLAR radiative transfer model for clear skies" submitted to Geoscientific Model Development (GMD) by Victoria Eugenia Cachorro, Juan Carlos Antuña-Sánchez and Ángel Máximo de Frutos**

*General comments*

The paper presents the SSOLAR-GOA model, which is a spectral radiative transfer model for the solar radiation under clear skies. The model provides global, direct and diffuse irradiances at the surface. The model is rather simple, since it assumes the atmosphere is a single homogeneous (plane-parallel) layer – a mixed layer of molecules and aerosols. The paper describes all components of the model in a clear manner. In addition, the model code is well documented and easy to use with a nice graphical user interface. The SSOLAR-GOA irradiances are validated against those simulated with the radiative transfer package libRadtran as well as field measurements. The results of this comparison are thoroughly elucidated. The model generally shows a good agreement with libRadtran simulations and measurement data throughout the majority of the solar spectrum under presented clear-sky conditions.

However, my research focus is radiative transfer in the presence of clouds – therefore my principal concern lies in the general applicability of this clear-sky model. Clouds are the main atmospheric modulators of solar radiation and profoundly impact surface irradiance. The incorporation of cloudiness in radiation codes is well established and should be considered in the next stage of the SSOLAR-GOA model development.

Overall, the paper is well structured and written (although grammar should be improved at several places). I support it for publication in GMD after a few comments are addressed as outlined below.

*Specific comments*

1) You are assuming the model atmosphere is a single homogeneous layer of molecules and aerosols. This looks too simplistic to me and it imposes limitations on the range of model applications. Clouds are ubiquitous in the atmosphere and solar irradiance at the ground is highly affected by clouds (e.g., Wapler and Mayer, 2008; Wissmeier et al., 2013; Jakub and Mayer, 2015; Clack, 2017; Črnivec and Mayer, 2019). Moreover, broken cloud fields can even enhance the global (i.e., direct + diffuse) surface irradiance compared to that on a clear-sky day. I suggest that you incorporate cloudiness at least within your single-layer geometry (or even in a vertical 1-D geometry and thus additionally include vertical layering of the atmosphere).

Keep in mind that many studies demonstrated that further cloud characteristics such as cloud vertical overlap and cloud horizontal inhomogeneity strongly impact surface irradiance, whereby the first parametrizations of these effects were developed a long time ago (e.g. Cahalan et al., 1994; Shonk and Hogan, 2008). Some of them – like the methodology of Cahalan et al. (1994) to account for internal cloud inhomogeneity – are even extremely simple yet they offer a substantial improvement (perhaps interesting for your future model development).

2) Part of the inaccuracy of your model is stemming from the fact that you don't account for vertical variation of the atmosphere, which handicaps primarily the accurate computation of diffuse radiation (as you also indicate in Section 4). Furthermore, although three-dimensional (3-D) radiative transfer effects are mainly related to clouds, there are regions of increased horizontal variations of aerosol optical properties (e.g., in the vicinity of aerosol sources) or atmospheric gases, where your simple algorithm would fail even in clear skies – please point out these limitations.

3) Line 189: libRadtran user's guide, 2015: You should rather mention the latest version issued in 2020. Also in Line 534 as well as in the References – the url for the document provided within the References (http://www.libradtran.org/doc/libRadtran.pdf) already points to the version from 2020.

4) Line 254: The optical thickness related to Rayleigh scattering by atmospheric molecules should be denoted as $\tau_R$ and not $\tau_a$, since the latter already denotes aerosol optical thickness.

5) In Line 435 you state that you have selected three different extraterrestrial work files in your model. You also show results using these various data. Can you provide the extraterrestrial irradiance files based on Wehrli (1985) and Gueymard (2004) in the model data folder? In the current Zenodo repository I can only find data from Kurucz (1992).

6) Section 4: You demonstrate that your model generally shows a good agreement with libRadtran simulations and measured data especially for direct irradiance (thanks to the classic Beer-Lambert-Bouguer law), whereas diffuse component exhibits a somewhat larger bias. While this might not be a big issue under clear skies where the diffuse surface irradiance is relatively low and has only a minor contribution to the global irradiance, the situation could change considerably in the presence of (partial) cloudiness. The latter would block a significant amount of direct radiation (reducing direct radiation reaching the surface) and simultaneously generate an increased component of diffuse surface radiation. This would act to increase the bias of global irradiance – an issue to bear in mind while extending your model to incorporate cloud conditions (suggesting that a more comprehensive parameterization of scattering might be needed together with the multi-layering of the atmosphere).

7) The way you currently write units for spectral irradiance in the entire manuscript (within the text as well as on figures): W/m$^2$ nm and W/m$^2$ µm is not physically correct (the correct form would be for example W/m$^2$/nm). However, according to GMD policy units must be written exponentially: W m$^{-2}$ nm$^{-1}$ (see https://www.geoscientific-model-development.net/submission.html).
Please consider this also when writing units for other physical quantities.

8) Figure 6: Labels should be added for at least some of the lines.

9) Section Code and Data availability: Please provide also the surface irradiance data from field measurements (presented in Section 4.2) in a Zenodo repository (or other reliable repository) to enable scientific reproducibility. See paragraph Data Sets at:
https://www.geoscientific-model-development.net/submission.html

*Technical corrections: typing and language errors*

10) Ensure proper and consistent naming of libRadtran throughout the manuscript:
Lines 21, 23, 201, 203, 212 and 494: 'LibRadtran' should be 'libRadtran'.
Lines 500 and 759: 'Libradtran' should be 'libRadtran'.
I am not sure how to properly start a sentence with 'libRadtran', but I encounter both uncapitalized and capitalized versions in your manuscript (see Lines 185, 190 and 527). I would suggest you start sentences as follows: 'The libRadtran package...' or similarly.

11) Ensure proper and consistent naming of the SSolar-GOA model throughout the manuscript:
Lines 356 and 759: 'SSolar_GOA' should be 'SSolar-GOA'.
I also encountered 'SSolar-model' (Line 541) and 'SSolar model' ...

12) Some abbreviations are introduced multiple times. For example:

Line 165: Use simply the abbreviation RTE, since you have already defined it in Line 159: Radiative Transfer Equation (RTE).

Line 233: Use simply the abbreviated form 'the BLB law', since you have already defined the Beer-Lambert-Bouguer (BLB) in Line 159. Furthermore, the expression is misspelled in Line 734, since it contains 'Bouger' instead of 'Bouguer'.

13) I noticed several very long sentences, which should be split into shorter sentences, for example:

Line 699: 'Certainly, the spectrum at SZA=82.59 (m=7.3) represents an extreme situation with very low spectral irradiance values, which may be of interest for some applications, such as the determination of the amount of absorbing gas, but of little interest as a solar energy resource at middle latitudes, but not negligible in very low latitudes since there are a larger number of hours with this insolation.'

14) Some other language corrections (although I noticed additional grammar mistakes in multiple sentences, so please check grammar once more):

Line 164: '… is to separate...' should be '… to separate...'.

Lines 172 and 179: 'ETR' should be 'RTE'.

Line 209: 'determine' should be 'determines'.

Lines 212 and 574: Dot (full stop) should be added at the end of the sentence.

Line 220: 'one-dimension' should be 'one-dimensional'.

Line 448: '… their features on wavelength...'  does not sound fine to me.

Line 501: 'consider' should be 'considers'.

Line 521: 'increases' should be 'increase'.

Line 552: 'were' should be 'was'.

Line 565: An additional parenthesis is missing after (2008).

Line 567: 'provided' should be 'provide'.

Line 593: 'thermal stabilized' should be 'thermally stabilized'.

---

## Author Comment (AC1)

**Answer to Reviewer 1**

Answers to the reviewer CrVirac for the manuscript
"SSolar-GOA v1.0: a simple, fast, and accurate Spectral SOLAR radiative transfer model for clear skies" submitted to Geoscientific Model Development (GMD) by Victoria Eugenia Cachorro, Juan Carlos Antuña-Sánchez and Ángel Máximo de Frutos

The authors thank to the reviewer for the effort to review the manuscript and for its fruitful and detailed comments.

**General Comments**

The paper presents the SSOLAR-GOA model, which is a spectral radiative transfer model for the solar radiation under clear skies. The model provides global, direct and diffuse irradiances at the surface. The model is rather simple, since it assumes the atmosphere is a single homogeneous (plane-parallel) layer – a mixed layer of molecules and aerosols. The paper describes all components of the model in a clear manner. In addition, the model code is well documented and easy to use with a nice graphical user interface. The SSOLAR-GOA irradiances are validated against those simulated with the radiative transfer package libRadtran as well as field measurements. The results of this comparison are thoroughly elucidated. The model generally shows a good agreement with libRadtran simulations and measurement data throughout the majority of the solar spectrum under presented clear-sky conditions.
**However, my research focus is radiative transfer in the presence of clouds – therefore my principal concern lies in the general applicability of this clear-sky model. Clouds are the main atmospheric modulators of solar radiation and profoundly impact surface irradiance. The incorporation of cloudiness in radiation codes is well established and should be considered in the next stage of the SSOLAR-GOA model development.**
Overall, the paper is well structured and written (although grammar should be improved at several places). I support it for publication in GMD after a few comments are addressed as outlined below.

**Response:** We answer to the main concern of the reviewer. The main aim of this work is to deliver a simple spectral solar radiation model under clear skies to a broad users community, not familiar with radiative transfer theory and specially to people in solar energy applications or educational frameworks, thus the applicability of the model is well defined. Spectral solar radiation models based on physical fundamental are scarce in most of these communities, being the majority of models obtained by parameterizations or expressions based on experimental data. The objective of this paper is to fill the gap between the Radiative Transfer Codes and those more simple solar models, working as an intermediate stage between them. The same reviewer emphasizes the simplicity of the model, "a single layer and not include clouds". The model emphasizes the Ambarsumian´s methodology of "addition of layers" to get an effective layer where the transmittance is given by a very easy expression. The simplicity of the expression against the two fluxes methods existing in the bibliography is a merit to mention, together with the fact that this methodology is less known comparatively to the different two flux methods.

As recommended, our purpose in future works is to include a plane-parallel cloud layer giving an "effective cloud-layer" which can produce the same surface irradiance that multiple layers for cloudy cases. Also to compare the Ambarsumian method with other two fluxes methods, well known in the bibliography but with a most complicated expressions. First results indicate a different behavior in the range 350-450 nm between them, and further research should be carried out

**Specific Comments:**

***Answer to point 1.*** This point was discussed above. We mentioned that future versions of the SSolar-GOA radiation model can include a cloud layer, but the transformation of the first version of the model, proposed in this paper, in a multiple layer model distorts the main objective of this work. As we mentioned, potential

uses of the SSOLAR-GOA model are suitable for educational and solar energy applications, which do not need really the inclusion of cloud effect. We thanks the reviewer about the recommendation on the application of the method by Callahan et al. (1994) in future versions of the model. We agree this could be the physical bases if a cloud-layer is included in our model.

*Answer to point 2.* We agree with the reviewer, but the differences between our model and libRadtran model are not only due to vertical structure and multiple layers. The way to obtain the diffuse components of the two models (libRadtran and SSolar-GOA) is completely different and the limitations are already explained in the text of the manuscript.

In this context we have applied our model under strong aerosol conditions: a desert dust intrusion in the Canary Islands. The predicted by SSsolar-GOA and libRadtran and measured surface irradiances are shown in the next Figure (in this case with integrated values). As can be seen for global and direct irradiance components the comparison is very good: relative error below 5% despite considering constant values for alpha and beta along the day. The variations of these parameters were weak and the estimated irradiances match the experimental values. However, we are working about this subject to improve these results.

[Figure]

*Answer to point 3.* Corrected. We have updated the reference based on your suggestions.

*Answer to point 4.* Corrected the typo about Rayleigh optical depth of line 254.

**Answer to point 5.** The other two files of extraterrestrial irradiance have been added to Zenodo and they are available for calculations in SSolar-GOA model. Also as recommended one file of direct normal and global irradiances have added to ZENODO to enable reproducibility.

**Answer to point 6.** Thanks for the suggestion, this will be take into account in the future version of the model when added clouds. Partially cloudy cases are difficult to simulate by 1D models, but we are going to investigate about this topic for further versions of the model.

**Answer to point 7.** All the units have been corrected throughout the manuscript according GMD policy units.
**Answer to point 8.** Four labels about the SZA values have been added to Figure 6 as required.

**Answer to points 9-14, about Technical corrections: typing and language errors.**

Based on the reviewer's good suggestions, the minor grammar corrections have been done, as RTE for ETR, those for the uniformity of libRadtran and SSolar-GOA names, etc..

---

## Author Comment (AC2)

**Answer to Reviewer 2**

The authors thank to the reviewer for the effort to review the manuscript and for its detailed comments.

**Reviewer comments**

The authors present a "simple, fast, and accurate" hyperspectral solar radiative transfer model for clear skies (SSolar-GOA v1.0). They evaluate the model against a state-of-the art radiative transfer model (libRadtran) and observations, showing an impressive accuracy and promising applicability in a multiple of different disciplines. ***Although, the overall analysis, focus, and results are, to an extent, appropriate for Geoscientific Model Development, as well as novel and important, I found the writing, organization, and presentation of results severely lacking. I would recommend thorough revising before further consideration***. I provide more detailed comments below, however I do think considerable revision is needed before a proper evaluation can be completed

PRIMARY COMMENTS
1. As stated throughout, I found the writing rather awkward, poor, or extremely confusing in several areas. This makes it challenging to follow the rationale, results, and discussion. Please consider a careful review of the writing with extra attention paid to sections/sentences that are awkwardly written.

2. The presentation of results focuses are merely visual or limited to percentual differences. A lot can be learnt from linear fits, and their r2's and RMSE's values. See specific comments for more context.

3. The model seems to do a great job, but the paper would be more interesting if the authors could explore limitations of the model as well, and move faster to results and discussions. Maybe add more discussion describing how the model could be improved, what areas are lacking, what type of simulations and scientific questions cannot be explored with this model, and how other areas could potentially benefit from this. I know the authors refer to other studies, but without really giving any concrete example in the paper. This is a real breakthrough and I can indicate a few:

Yang, P., Prikaziuk, E., Verhoef, W. and Van Der Tol, C.: SCOPE 2.0: a model to simulate vegetated land surface fluxes and satellite signals, Geosci. Model Dev, 14, 4697–4712, doi:10.5194/gmd-14-4697-2021, 2021.

Braghiere, R. K., Wang, Y., Doughty, R., Sousa, D., Magney, T., Widlowski, J.-L., Longo, M., Bloom, A. A., Worden, J., Gentine, P. and Frankenberg, C.: Accounting for canopy structure improves hyperspectral radiative transfer and sun-induced chlorophyll fluorescence representations in a new generation Earth System model, Remote Sens. Environ., 261, 112497, doi:10.1016/j.rse.2021.112497, 2021.

**Response to general and primary comments by the authors. To do this we have numbered the paragraph**

1. We have followed most of the reviewer recommendations in order to improve the manuscript. Regarding the writing, we try to improve it, but we also think that the editorial can improve it in the article as a whole. Certainly the writing is not as good as we wish. We are not English natives but the text was proofread by a professional English translator and in our opinion it follows the normal rules of English language and the sentences commonly used in our research field.

2. We have evaluated the statistical RMSE% and the parameters of the linear regression for the comparison in tables I and II. These values are also discussed in the analysis of results in the new version of the manuscript. We consider that figures of the fits do not report more significant information, hence they are not included since they greatly lengthen the manuscript. However, we show some of them here below.

3. We have changed the paragraph "Conclusions" for a new one called "Discussion and conclusions", where we emphasize the limitations and advantages of the model, how the model could be improved or in which areas it can or cannot be applied as the reviewer recommends.

**Abstract**

The abstract is too long and contains some methodology. The abstract should concise and describe general relevance and main results. Line 12-18 could be removed. Starting the abstract with the general applicability of the study may attract interest. This section should be re-structured.

Response: The abstract has been shortened and restructured. However, the main characteristics of the physical methodology must be clearly explicit: it is the core of the model and defines the model with respect to other models that are based on the two flux methodology. As we mentioned in the text, the model tries to fill the gap between the detailed-complicated RT Codes and the most simple parameterized solar radiation models (mostly based on experimental data).

Line 10: are adapted? It looks like something is missing. It looks like it is a direct translation.
Response: sorry this is an error where "are" is "and", but this sentence has been removed in the new manuscript version.

Line 14: "sufficient accuracy" – can you provide an $r^2$? A RMSE in percentage? Anything that exemplifies what that means.
Response: This statistical indicator has been added and evaluated in the new Tables I and II.

Line 28: Avoid wording like "obviously" in scientific writing.
Response: Yes, it was removed.

**Introduction.**
Line 32: Earth-atmosphere System
Response: done

Line 36- energy?
Response: replace by "solar energy"

Etc is a vague word and should probably be used minimally.
Line 45 – what is etc? be precise. Please define the spectral wavelengths associated with UV, visible, etc.
Response: "etc." has been removed. The spectral ranges have been clarified and added as: *"(i.e., UV (~300-400 nm), visible (~400-700 nm), near-infrared (~700-1000 nm), entire solar range (~300-3000 nm))"*.

Line 55 – do not refer other studies in this way. Just write these between brackets.
Response: Done

Line 60 – etc.
Response: it has been removed.

Line 71 – 1-10 nm is low to medium? Don't you mean medium to high?
Response: we consider "high" below 1 nm. Most of detailed RT models for atmospheric science applications works with a spectral resolution below 1 nm.  See that in RT Theory most if the classical books start with gas molecular absorption, and hence with the concept of "line absorption" and its parameters, like position, intensity and half-width and hence the line-by-line models are recommended for many applications. Considering this type of RT Codes, to work with 1 nm is consider a wide interval where thousands of spectral lines are included, but it also depends on whether we are in the UV at 0.3 μm or in the far-infrared about 15

µm. On the other extreme are the RT models used by climate models, where solar range is taken with 1 or 6 intervals as maximum and hence the K-distribution is currently applied. In satellite remote sensing applications, the term "hyperspectral" is considered as a high spectral resolution but this is relative. Currently sensor satellite remote sensing applied to vegetation used less spectral resolution that those use for the atmospheric component determination. However, to say low, medium or high is "relative", in general depends on the context you are working or speaking and it will depend on each specific area of work.

Line 79 – libRadtran reference?
Response: Done

**Material and methodology.**
This sections is way too long and could be substantially reduced, with some of the sections moved into a Supplementary material or appendix.
Response: The last paragraph has been removed and sent to section 4.2. To add supplementary material or appendix enlarged the article.

Line 139 – etc.
Response: removed.

Line 142 – Earth
Response: done.

Line 159 – the BLB law.
Response: done.

Line 160 – which component?
Response: Done, the sentence was modified as "only to direct component.

Line 161 – This gives rise? What does that mean?
Response: this sentence has been replaced by "This allows".

Line 163 – etc. Paragraph 3.1?
Response: done.

Line 164 – there are two verbs in this sentence.
Response: yes, the verb "is" has been removed.

Line 165 – you already defined RTE before.
Response: yes, thank, we only put RTE.

Line 166 – to solve -> solving
Response: done.

Line 168 – specific problem involved? This is so general. Give examples
Response: yes, it is so general but it fits the phrase where it is included, we do not believe any further clarification, we refer to the books where the specific problems are solved.

Line 172 – ETR?!
Response: all "ETR" have been replaced by "RTE".

Line 173 – for the diffuse component only.
Response. done, we also have removed the parenthesis after global component in this sentence.

Line 174 – Not only to the atmosphere, but adapted for canopies to:
Sellers, P. J.: Canopy reflectance, photosynthesis and transpiration., Int. J. Remote Sens., 6(8), 1335–1372, doi:10.1080/01431168508948283, 1985.
Response: yes, it is true, certainly the methods for solving the RTE can be used or applied to atmosphere and vegetation studies and the SSolar-GOA model may serve as input for vegetation transfer models at the canopy level, as SAIL, SCOPE and others, providing spectral solar irradiances at the top of the canopy. Bear in mind that our main area of research is the atmosphere but vegetation radiative transfer models are also familiar in our research group (see the reference Berjón et al. (2013)). Many thanks for these two recent references. We have tried to incorporate this information in the discussion section.

Line 179 – ETR?
Response: done.

Line 197 – BLB law.
Response: done.

Line 212 – period missing.
Response: done

Line 224 – Again, 1-10 nm is a very resolution.
Response: it has been discussed above.

Line 226 – what is this error?
Response: about 2-5%, this information has been added in the text.

Line 231 – Thank you for giving the link to the model. How can the direct component be higher than the global one for some wavelengths?
Response. as can be seen in Figure 1b and c, for normal input parameters as those of the figure but for SZA higher that 30 degrees, direct normal component is higher that global but not the horizontal component.

Line 233 – You already defined BLB.
Response: done.

Line 289 – Use the symbol of micrometers.
Response: done.

Line 321 – 1 DU instead of 1 Dobson.
Response: done.

Section 3.3. This list of items could be a Table.
Response: Yes, but it is an option and not relevant since there is not so much information.

**Results.**
Fig1. Add degrees to the numbers next to SZA. Write down Direct-horizontal instead of **dir-how**. Figures should be directly interpretable.
Response: done.

Line 444 – Before the comparison? What?
Response: We have replaced the sentence by "Before the comparison between both models",

Fig 1 and 2 could be combined into one single figure, with the top row being fig 1 and bottom row fig 2. Ozone = 300 DU, not Dobson. Add units of all the other parameters too.
Response: to join Figure 1 and 2 is not convenient since they give different information. Figure 1 gives a general idea about the values of the three component and their variation with the SZA. Figure 2 is related or

equivalent to figures 4 and 5, giving direct normal, global and diffuse information about the comparison with libRadtran, therefore we think joining figure 1 and 2 is not convenient. Ozone unit as DU has been added. The water vapor is the only with units as already it appears (as cm) and the other are dimensionless.

Fig3 should include SZA= 6 deg as well. Be consistent.
Response: Figure 3 is not equivalent to figures 2, 4 and 5. This is the reason why we don't draw the corresponding 6° or 60°. This Figure 3 is shown to emphasize the different spectral resolution between the libRadtran and SSolar-GOA models as revealed by the absorption of water vapor bands, giving rise to the high differences observed as both positive and negative peaks.

Fig 4 is repetitive and could probably be moved into supplementary material.
Response: we consider that Figure 4 is not repetitive, it is consistent with Figure 2 and 5.

Line 533 - see libRadtran user Ì□s guide, 2015? Please reference appropriately.
Response: done. The reference is already given above.

Fig 4 and 5 could be combined into a single one too. Same thing about adding degrees next to the SZA numbers throughout.
Response: we have explained the consistence of figures 2, 4 and 5. Degrees have been added in all the figures and text.

Fig 6 – what are the different colors? Please use an include color scheme suitable for colorblind people.
Response: we have added the values of four SZAs as required by reviewer 1, and the symbol of degrees to SZAs.

Fig 7 – Please add the runs from libRadtran here for comparison too.
Response: this has been discussed above.

Line 586 – How do you know the agreement is "excellent"? Visually, it looks great, but could add some statistics into your evaluations? A linear fit with observed/simulated with libRadtran versus SSolar-GOA ($r_2$, RMSE, and slope) could tell us so much about model performance.
Response: done.

Fig 8 - Please add the runs from libRadtran here for comparison too.
Response: we have dedicated the first part of the article to this comparison with libRadtran. We think that the addition of the modelled data by libRadtran to the measured data is confuse for this figure. Our purpose here is validating the SSolar-GOA model con experimental data.

Fig 9 – This is not your work, could probably be moved into supplementary material. Please add the full citations in the figure, e.g., Kurucz, 1992.
Response: Yes, it is a possibility, but we prefer to present figure 9, since these differences between the values of the extraterrestrial irradiances are very important when analyzing the absolute and relative differences in the comparison between experimental and modeled solar radiation spectra. Although this is well known, the values of these differences must be remembered (as it is illustrated in the figure) when making the comparison between modelled and measured spectra. Citations have been added to the Figure as required.

Line 634 – add comma after 'To this'.
Response: done.

Fig 10 – show linear fit with $r_2$ and RMSE.
What is the purpose of Fig 11?

Response: We have added Tables I and II for the earlier figures 7 and 8; we consider that all information is collected in these tables. Figures 10 and 12 are not illustrated for comparison objective, but they want to emphasize the different capabilities of the ASD compared with other spectroradiometers: its largest spectral range from 400 to 2200 nm (thus, loosing information in the UV range) and is high time resolution, which can be of interest for other type of applications.

The purpose of figure 11 is to show that reliable AOD values are used as input in all modeled data in the comparison between modelled and measured solar spectra and that AOD is the main parameter in the comparison of solar irradiances under clear skies.

Fig 12 – show linear fit with $r_2$ and RMSE.
Response: it has been discussed above.

**Conclusion.**

Line 708 – avoid huge and extensive.
Response: done.

Line 711 – avoid enormous.
Response: done.

**Figures of linear fits**

[Figure]

[Figure]

SZA=60°

[Figure]

Day 16 July 2002,
Veleta Campaign

---

## Editor Decision (ED1)

Nina Črnivec's comments on a revised version of the manuscript entitled:

**"SSolar-GOA v1.0: a simple, fast, and accurate Spectral SOLAR radiative transfer model for clear skies"**
submitted to Geoscientific Model Development (GMD) by Victoria Eugenia Cachorro, Juan Carlos Antuña-Sánchez and Ángel Máximo de Frutos

In my opinion, the quality and the extent of the revised manuscript is adequate for GMD. I thank the authors for making the changes and thereby improving the initial article. I support the publication in GMD as elucidated below.

The majority of comments provided in the first round of review have been properly addressed. The cloudiness has not yet been incorporated in the radiation model, but it is fine if it will be included in a future model version. For now, I appreciate the accuracy of this simple model in cloud-free atmospheric conditions when compared to more sophisticated libRadtran benchmark calculations. I can see the value of this simple model for clear skies to serve as an educational resource for researchers that are entering the field of radiative transfer.

The Zenodo repository has been updated with missing data files so that scientific reproducibility is ensured. Last but not least, the critical grammar mistakes have been corrected.

---

## Author Response (AR2)

**Dear Topical Editor Sylvester Arabas**

First of all, let me point out that the GMD code availability policy extends to all code needed to reproduce results presented in the paper including scripts that automate model runs and visualization. In the case of the present paper, this should cover scripts automating the SSolar_GOA and libRadtran runs for the comparative analysis. Please provide the code required to reproduce the plots presented in the paper (e.g. as an electronic supplement to the paper or at a persistent repository as Zenodo).

*Answer: We have inserted in Zenodo the input file to run the libRadtran. Related with the plots, each figure was created individually with Excell 2013 and we do not understand what means to provide the code.*

**Answers to the comments related to the minor corrections to the revised manuscript "SSolar-GOA v1.0: a simple, fast, and accurate Spectral SOLAR radiative transfer model for clear skies" submitted to Geoscientific Model Development (GMD) by Victoria Eugenia Cachorro, Juan Carlos Antuña-Sánchez and Ángel Máximo de Frutos**

**The authors thank to Editor for the effort to review and the supervise the manuscript**

* Page 1 (abstract)
- line 7: physical-based -> physically-based----------------------------- *Answer: done*
- line 17: sentence "Besides, ..." unclear ---------------------------*Answer: it was removed*
- line 18: remove "extensive" ------------------------------------------- *Answer: done*
- line 21: "shows a high performance ... and it underestimate" - please shorten, perhaps clarifying that the presented validation quantified the relative differences...
*Answer: The earlier sentence " From the results of the comparison with libRadtran, the SSolar-GOA model shows a high performance for the whole spectral range and" was changed to "The SSolar-GOA is validated by a quantitative comparison with libRadtran, showing that "*
- line 27: remove "Obviously"-------------------------------------------------------- *done*
- given the journal and paper scope, I suggest to include in the abstract and introduction a mention of implementation in Python and open-source licensing
*Answer: two short sentences was added, one to the abstract (line 30-31) and one to the introduction (line 78) according your recommendation.*

* Page 2:
- line 48: should "Research Centres" be capitalised?-------------- *Answer: no, changed*
- line 51: crossed out "in a"    ----------------------------------------- *Answer: done*
- line 52: Hodges 1993 cited as Hodges 1990------------------- *Answer: changed to 1993*
In the introduction, related with the refrences to Data bases is not easy how to be

* Page 3:
- line 93: "allows" -> "allow" ------------------------------------------- *Answer: done*

* Page 4:
- line 116: "has the drawback" -> "comes with a tradeoff" ? -----
*Answer: This sentence " The ASD solar irradiance….. " has changes substantially (lines 115-118)*

* Page 5:
- line 131: the "or" in units parenthesis is unclear ----------------------- *Answer: removed*
- line 134: "widest used" -> "most widely used" ---------------------------- *Answer: done*
- line 145: rephrase "disappeared" ----------------------------------------------
*Answer: the sentence was changed to "*As can be seen, the explicit dependence on the cos(SZA) of expression (1) is removed in expression (2)"*.*
- line 153: "law" missing after Bouguer --------------------------------------- *Answer: done*
- line 153: remove "easy" ----------------------------------------------------- *Answer: done*

* Page 6:
- line 163: "solver" -> "solvers" ……………………………………… *Answer: done*
- line 166/167: "but contains" -> "and comes in" ? ----------------- *Answer: changed to "and presents"*
- line 178: "which has undergone..." - what is the purpose of this statement?.
*Answer: the idea was to explain the origin and evolution from a program "uvspec" to a big Code and now a package that contains many Codes as a whole. We added at the end of the sentence "*……………… to reach the current libRadtran estructure"*
- line 181: package -> source code ------------------------------------------ *Answer: done*
- line 184: remove "really" ---------------------------------------------------- *Answer: done*
- line 186: two sentences in a row begin with "Therefore, "-------------- *Answer: done*
- line 187: what does "the Mie program" refer to?------------------
*Answer: clarified, we have added "*(see libRadtran user´s guide Chapter 4) "the calculation" into the total sentence (lines 190-192)

* Page 7:
- line 201: "high" -> "height" ? -------------------------------------------------------- *Done*
- line 205: "though" -> "through" ----------------------------------------------------- *Done*
- line 206: give example references to these models-------------------------- *Done*
- lines 207-209: "This is a physical, fast, ..." suggest removing this sentence--*Done*
- line 209: "As already mentioned, the core of the model is the simplicity of the" -> "The crux of the model is the simple" ---------------------------------------------- *Done*
- line 251: rephrase "working very well" (aimed at operation within the ... range ?)
*Answer: done*
- line 220-221: "The model, in some way, ..." - unclear sentence
*Answer: the sentence was changed to "*The model may be easily adapted to the case of limited available information about model's input parameters".*
- line 221: "The model as described may be easily replicated by the readers, or it may be download Windows version"
-> "The SSolar_GOA v1.0 is released as free and open-source software.
It is implemented in Python offering portability across architectures and operating systems.
For download instructions, see the Code Availability section."
- *Answer: Your recommended sentence have been added in the text and removed that of the earlier version (lines 227-230)*

* Page 8:
- line 235: "and various" -> unneeded "and" ------------------------------------------ *Done*
- line 140: "where implicitly it is assumed the non-interaction between these processes" -> "where the non-interaction between these processes is implicitly assumed" ----------------------------------------------------------------------------- *changed*

- equation 7: looking at the code, the exponent at the last lambda in the denominator should be "-4" not "4"  ------------------------------------
*Answer: It was an error, is -2 and expression (7) was corrected in the manuscript. This error do not make any influence in the values of output irradiances of the model*
- line 270: "and sea level" -> "and the sea-level pressure" ----------------------- *replaced*

**\* Page 9:**
- line 296: "model includes a file": this mixes implementation with formulation, would better sounds as "model uses tabulated coefficients" ? ------------------------ *Yes, done, changing de sentence*

**\* Page 10:**
- line 304: remove "very"---------------------------------------------------- *Done*

**\* Page 11:**
- line 338: "However, inverse to" -> "In contrast to"---------------------------- *Done*
- line 342: suggest removing "so care must be taken with the units of both quantities in the previous expressions" ----------------------------------------------*Done*

**\* Page 12:**
- line 373: please rephrase "as necessary approaches for developing a simple model under the consideration of non-interaction"
*Answer: Sorry, It is not easy to modify this sentence. However, it was modified. We try to say that scattering and absorption processes are taken without interaction between them which simplify considerably the formulation of the model. The new sentence is "*Scattering and gas absorption are applied to a single atmospheric homogeneous layer in the SSolar-GOA under the consideration of non-interaction of both processes, which simplify considerably the formulation of the model."

**\* Page 13:**
- line 380-381: remove "but a spectral file for ρ is very easy to implement in the model"  -------------------------------------------------------------------- *Yes, removed*
- line 393: "are usually taken from the bibliography." - please rephrase and be precise (what "usually" and "from the bibliography" mean?)
*Answer: done, changing a few the sentence and adding two references "*given in different publications (Dubovik et al., 2004; Hamill et al., 2016)".
- line 393/394: remove "Finally, we call attention to the total number of expressions/formulas..."
(or rephrase and elaborate being precise which other models you refer to and what "number of formulas" implies: implementation challenges, computational cost, ..)
*Answer: done, changing a few the sentence and adding tree references "(*e.g.: Bird, 1984; Gueymard, 1995, 2005; Xie and Sengupta, 2018)"
- line 395: suggest adding a screenshot of the graphical user interface here
*Answer: done as Figure 1 and all figures were renumbered. As consequence the last sentence of section 3 was changed to "*In our model, we it can select three different extraterrestrial work files, given by Wehrli, (1985), Kurucz, (1992) and Gueymard, (2004), as it appears in Figure 1."

* Page 14:
- line 415: remove "bear in mind that" ------------------------------------------------ *Done*
- line 433: remove "Before the two comparisons,"------------------------------------ *Done*

* Page 18:
- line 489: rephrase "interval in nm of the model"-----------------------------
*Answer: the sentence was slightly modified and "in nm" was removed. I hope this results in a more clear sentence*
- line 501: "it can observe" -> "one can observe" ? --------------------------------- *Done*

*Modification in Page 19-20: The paragraph just after current Figure 9 was moved before the Figure.*

* Page 24:
- line 579: "consistence" -> "consistency"   ----------------------------------------- *Done*
- line 581: "component are" -> "components are"   --------------------------------- *Done*
- line 583: "range stand out" -> "range stands out"  --------------------------------- *Done*

* Page 27:
- first paragraph of 4.2 is all bold---------------------------------------- *Yes, it was corrected*

* Page 39:
- line 785: remove "it is true that"  ----------------------------------------------- *Removed*
- line 795: rephrase "easy-to-understand"---------------------------------------------
 *Answer: now at line 794. The sentence* "built with a set of easy-to-understand input parameters" *was slightly changed to* "based on a set of input parameters easy to use and understand."

* Page 40:
- line 807: "and the idea is to implement this new" - rephrase, perhaps like "will be considered in further model development" (to clarify it is not available as of now
*Answer: now at line 805-808, senetence modified as recommended*
- line 813: rephrase around "two more influent" (and remove parenthesis)

*Answer: Changed to "* being these two atmospheric components the most influential".

- line 815: "is easy to understand and evaluate" -> "results in concise and computationally undemanding formulation" ? -------------------------- *replaced*
- line 815: "but more important is that" -> ". Importantly, it was shown that the assumption of" ----------------------------------------------------------------------- *Done*
- line 818: rephrase "one layer or multiple layer do not added"------------- *Done*
- line 822-823: correct "for many applications in solar energy for different application agriculture" ------------------------------------- *Done, with some modifications*
- line 824: "Depending of the required accuracy" -> "Depending on the required level of accuracy"------------------------------------------------------------------- *Done*

* Page 41:
- line 831: correct "which is covers" -------------------------*corrected to* "*which is covered"*

- line 836: rephrase "but it must be in mind", "easy to use, which cannot to compete" --------------------------------------------------------- *the sentence was changed to* "which cannot to compete with multilayer RT Codes that solve the RT equation"
- line 837: rephrase "It is not easy to find in" to "To the authors' knowledge"-- *Done*
- line 839: correct reference year (1919)-------------------------------------- *Corrected*

\* Code availability:
- Please elaborate what you mean by "portable". Given the code is developed in Python, it is portable across platforms and operating systems by design.
*Answer: We replace the sentence: "A portable version can be downloaded for Windows users…" by "For Windows users a binary package has been generated which can be downloaded from.."*

\* Zenodo archive:
- the readme file mentions installation, but it's more a "installation of dependencies" rather than package installation.

*Answer: The installation instructions include the installation of the dependencies because this is the only thing necessary to run the application.*

- the archive contains numerous cache files (__pycache__ dirs with *.pyc files)
*Answer: done*
- the archive is a tarbomb, please include a top directory in it
*Answer: done*
- the 1.7MB size of the .ico file is intriguing
*Answer: done*
- after launching the program for the first time, the parameter values do not match defaults (clicking "Load default inputs" on a freshly opened window changes many parameters)
*Answer: done*
- the software license is not specified within any of the files in the code archive, please at least include a COPYING file
*Answer: We add LICENSE file*

\* Figures:
- all figures need to be supplied in a vector graphics format (not raster graphics/screenshots)
(similarly, in the SSolar_GOA user interface, the "Save image" feature would best allow to save in publication-ready vector format (svg, pdf, ...)
*Answer: Vector graphics format was already done in the previous version. The "Save image" of SSolar-GOA user interface is in svg format.*

\* References (see https://www.geoscientific-model-development.net/submission.html#references):
- some entries have journal names abbreviated, some not (see the above URL for suggested abbreviation database),
*Answer: the journal abbreviation was revised and write correctly*
- some abbreviations include dots, some not,
*Answer: we had remove the dots, but hey appears in the original article-reference*
- some entries use all-words-capitalised, some first-word-capitalised format,

*Answer: This was revised and all entries use capitalized format*

- Fouquart & Bonnel: missing capital letter (Earth's)---------------------------- *done*

- libRadtran user's guide: multiple years unclear, missing authors (Bernhard Mayer, Arve Kylling, Claudia Emde, Robert Buras,Ulrich Hamann, Josef Gasteiger, and Bettina Richter), missing version number, please consider asking the authors to post the pdf at a permanent location (arxiv, zenodo, ...)

*Answer: We put two references separated, 2015 and 2020. To ask the authors requires more time.*

- ASD Full Range: wouldn't "Malvern Panalytical, 2021" better serve as label?

*Answer: We prefer to put ASD, because this instrument is known as ASD into the scientific community.*

- Hodges 1993: unclear what the 1993 year refers to, the website pointed by the DOI does not list Hodges name, "Unitet State" – typo

*Aswer: This reference is complicated because is not a journal article.We have modified the references according to https://www.osti.gov/biblio/108148.*

*Answer:*

- Kurucz 1992: is it this paper: https://doi.org/10.1017/S0074180900124805 (if so, the year is invalid)

*Answer: the doi was corrected and the year is 1992, Observe that the reference has been changed substantially.*

- Sukhodolov 2014: bogus doi url, missing capital letter in "evaluation"

*Answer: done*

- Utrillas and Vergaz entries are coalesced

*Answer: done*